# STYLIST: STYLE-DRIVEN FEATURE RANKING FOR ROBUST NOVELTY DETECTION

## ABSTRACT

Novelty detection aims at finding samples that differ in some form from the distribution of seen samples. But not all changes are created equal. Data can suffer a multitude of distribution shifts, and we might want to detect only some types of relevant changes. Similar to works in out-of-distribution generalization, we propose to use the formalization of separating into semantic or content changes, that are relevant to our task, and style changes, that are irrelevant. Within this formalization, we define the **robust novelty detection** as the task of finding semantic changes while being robust to style distributional shifts. Leveraging pretrained, large-scale model representations, we introduce **Stylist**, a novel method that focuses on dropping environment-biased features. First, we compute a per-feature score based on the feature distribution distances between environments. Next, we show that our selection manages to remove features responsible for spurious correlations and improve novelty detection performance. For evaluation, we adapt domain generalization datasets to our task and analyze the methods' behaviors. We additionally built a large synthetic dataset where we have control over the spurious correlations degree. We prove that our selection mechanism improves novelty detection algorithms across multiple datasets, containing both stylistic and content shifts.

## 1 INTRODUCTION

In the wider body of literature, Novelty Detection (**ND**) (Li et al., 2023a; Pimentel et al., 2014; Salehi et al., 2022; Tack et al., 2020; Ruff et al., 2021; Yang et al., 2021) has conventionally revolved around the identification of notable and meaningful deviations from established data distributions. The ND task is often used interchangeably with the broader anomaly detection task, but there is a noteworthy difference between the two. Anomalies are fundamentally distinct from typical samples and can manifest as deviations in various forms. Novelties, or semantic anomalies, represent a subset of anomalies, specifically targeting semantic deviations, aiming to identify any test sample that does not conform to the established training categories. For instance, in practical scenarios such as medical diagnosis (Chauhan & Vig, 2015), financial fraud detection (Bhattacharyya et al., 2011) or network intrusion detection (Dragoi et al., 2022), the primary objective is to detect novelties, such as unique aspects of a cell's biological structure, while disregarding irrelevant divergent characteristics, such as artifacts stemming from equipment.

Our main point is that not all changes are created equal. When we move across a continent using a self-driving car, we might be amazed by the style of different houses that we haven't seen before, but the self-driving car should still behave the same. On the other hand, when encountering a new structure that it hasn't seen before, such as a new type of intersection or bridge, the car should *detect* that this is a *novel* situation and cease the control to the driver.

Thus, we define **semantic** or **content** shifts as the changes in data distribution that involve some factors that are relevant to our task (such as driving), and **style** shifts as the changes that involve some factors that are irrelevant to our task. Many times, the style factors are correlated with content factors, so when learning the semantics of a problem, we might learn some spurious correlations involving irrelevant style factors. These spurious correlations might not always hold, thus we should not rely on them.

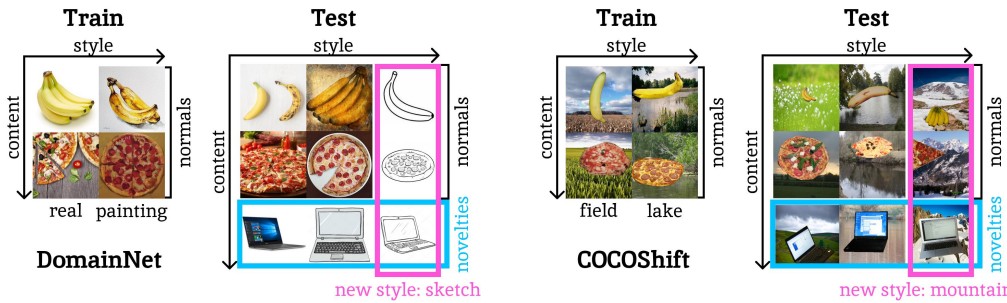

Figure 1: Multi-environment setup for the Robust Novelty Detection task.

In this context, we focus on **robust novelty detection**, which aims to identify distribution changes in content, while ignoring style changes.

To be able to distinguish between the two, we consider the multi-environment setup from the distribution shift studies (Koh et al., 2021; Zhou et al., 2022), where besides the usual content label, we also have access to a style label. An environment is composed of samples with a particular style category, but having any content categories. In this scenario, a style category is basically a set of factors or relations that hold only in one environment (*e.g.* for the self-driving car example, driving in the forest or near a beach, or even in some fictional, Disney like scenario can be seen as different styles). On the other hand, a content category refers to a set of factors or relations that hold across all environments (*e.g.* the roads, cars, bikes, humans categories). The style component characterizes the data in an uncertain, maybe even spurious way, toward the content classification task. During training, the content may be correlated with other factors from the training environments, which are irrelevant to this new task and might become spurious. This is a challenging problem for content classification tasks and even more challenging in the novelty detection setup, where during training you only observe a set of known classes.

With this in mind, our work centers on detecting novel content, while removing environment-biased features. Specifically, we propose a method to rank features based on their distribution changes across training environments. This ranking mechanism, followed by dropping environment-biased features, aims to enhance the performance of novelty detection methods, enabling them to generalize more effectively in the presence of spurious correlations and to give us a glimpse of features' interpretability.

Summarized, our **main contributions** are the following:

- We propose **Stylist**, a simple, yet very effective algorithm that scores pretrained features, based on their distribution changes between training environments. We prove that this approach ranks features based on how much they are focused on the environmental details and gives a glimpse of interpretability to the "black-box" embeddings.
- We show that, by gradually removing the environment-biased features proposed by our algorithm, we significantly improve the ND models' generalization capabilities, both in the covariate and sub-population shift setups, by up to 8%, with the benefits being more pronounced at a high degree of spuriousness correlations.
- We introduce **COCOShift**, a comprehensive, synthetic benchmark, with 4 levels of spuriousness, which enables a detailed analysis for the Robust Novelty Detection task. We also adapt the DomainNet and fMoW multi-environment real datasets to novelty detection and validate our main results in this setting.

## 2 PROBLEM FORMULATION

Real-world data suffers from a multitude of changes that we usually refer to as distributional shifts. As described by Smeu et al. (2022) these kinds of shifts are involved in different lines of work, with different goals: domain generalization wants to be *robust to style shifts* while most anomaly detection methods want to *detect either style or semantic shifts*. We denote robust novelty detection

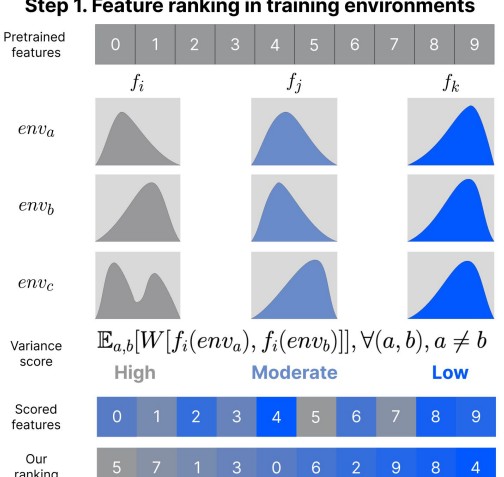 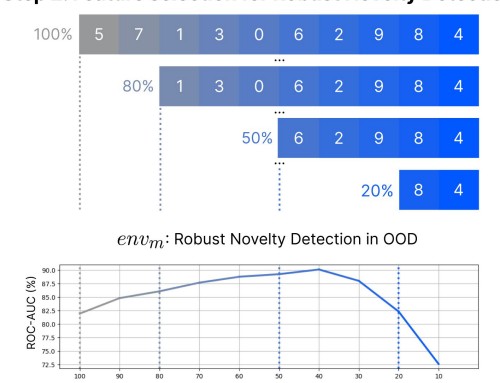

Figure 2: **Stylist**. We improve the ND performance by identifying (Step 1) and gradually removing (Step 2) environment-biased features. From this point of view, higher distribution distances between environments proved to be a good indicator for ranking features.

as the task of detecting semantic novelties while being robust to style distributional shifts. More exactly, detect samples that differ by some semantic shifts from some *seen* training samples, while ignoring samples that are only affected by style shifts.

We work in a multi-environment setup, where each training environment changes the style of the samples, while all the environments contain a set of *seen* content classes. The goal of training environments is to define what is content and what is style. Consequently, we are not restricted to a certain style definition but rather expect the training environments to define what might be correlated with our task, but is not actually relevant. Then, we define an evaluation environment, containing *both seen and novel* classes with an associated new style. The goal of **robust novelty detection** is to separate between *seen* and *novel* content classes, without being affected by the new style.

We focus on multi-class novelty detection, where we have training environments with multiple content classes. However, we treat them as a single group of normal samples and ignore their content labels. By the definition of novelty detection task, there is a zero level of corruption among the normal samples, as opposed to the more common setup of anomaly detection.

In Fig. 1 we present two scenarios to exemplify our setup. In the first example, the normal samples encompass representations of objects in various formats (real images or paintings). In this context, style is defined as the manner of depiction. During testing, our objective is to correctly categorize the laptop as a novel class. Furthermore, we must discern that the sketch of the banana, despite the shift in style (from real images and paintings to sketches), is not a novel class. In the second example, we observe a different definition of style, namely the background of the images, which should also be irrelevant for classifying the content.

## 3 OUR APPROACH

Some dimensions of a given pretrained representation could be more representative of the semantic aspects while others might be more representative of style elements. To minimize the impact of style factors on our novelty detection task, we aim to reduce our reliance on them. Thus, it might be that we are better off ignoring the dimensions that mostly contain style information, which we denote as environment-biased features. We focus on discovering which features from a given, pretrained representation, are more environment-biased, thus prone to contain spurious correlations, and should be better ignored. Finding the robust part of a representation is closely linked to invariance between environments, thus we want to have a measure of variance for each dimension in our representation. We first quantify the degree of change in each feature distribution, and then we drop the ones that vary more, as depicted in Fig. 2.

Our algorithm works over the representations extracted from a frozen model. We assume that for each sample of our training set, we start with a vector of $N$ features. We proceed in two steps:

**Step 1. Feature ranking in training environments** First, we compute a score that says how much a feature changes across environments. For each feature $i$, we consider $f_i(env)$ to be the distribution of this feature in environment $env$. We employ the Wasserstein distance to compute the distance between the distributions of each feature $i$, across pairs of environments $(a, b)$.

$$dist_i(env_a, env_b) = W(f_i(env_a), f_i(env_b)), \quad \forall i \in [1..N] \tag{1}$$

The per-feature score is obtained next as the expected value of the Wasserstein distance across all pairs of environments $(env_a, env_b)$, where $a \neq b$.

$$score_i = \mathbb{E}_{a,b}[dist_i(env_a, env_b)], \quad \forall i \in [1..N] \tag{2}$$

**Step 2. Features selection for Robust Novelty Detection** Next, since our purpose is to be robust and to be able to ignore environment changes, we remove features with top scores. The intuition here is that environment-biased features facilitate spuriousness, providing a training setup prone to such correlations.

The exact used distance might not be that important, but what matters is the process of looking at differences between environments and searching for what consistently changes between them (*e.g.* in terms of distribution). For this, in our approach, we rely on the following assumptions:

1. The pretrained feature extractor is able to represent both seen and novel content categories, as well as known and new styles.
2. The change in style dominates over the change in content when we compare between different environments.

First, our approach leverages pretrained embeddings with extensive coverage across various content and style categories. This eliminates the need for employing domain adaptation techniques on the pretrained representation, enabling us to keep the model frozen and exclusively utilize its features. To illustrate this concept, consider the initial scenario presented in Fig. 1, where we presume that the pretrained model effectively captures pertinent features for objects like bananas, pizza, and laptops across diverse depictions such as real images, paintings, and sketches. This assumption is easily met in practice nowadays when we have access to powerful pretrained models that have been trained on large and comprehensive datasets. So the difficulty does not lie in getting good representation, but at the next level, where, given a set of features, you need to select the ones that are relevant for identifying novel content while dropping style-related features that can cause spurious correlations.

As for the second assumption, in our setup, both style and content can vary across environments. Since random changes would create an impossible problem, we assume that style-induced changes in the data distribution are greater than content ones, when we look at two different environments. This implies that our environments have different associated styles, and the considered representation can capture these relevant differences, which is covered by our first assumption. In order to mitigate these challenges, it is imperative that we operate in a well-constructed multi-environment setup to be able to gain relevant information about the style shift. In the context of the Novelty Detection task, where both style and content distributions can shift, the quality and quantity of these environments play a pivotal role. They serve as reference points to delineate and clarify what holds significance and what does not within our task.

## 4 EXPERIMENTAL ANALYSIS

Our experimental analysis is conducted using two real datasets and a synthetic one. For the first two, we employ adaptations of well-established domain generalization datasets: fMoW (Christie et al., 2018) and DomainNet (Peng et al., 2019). All are multi-environment datasets and for each, we divide the environments into two sets denoted as follows: in-distribution (**ID**) environments (associated with styles that we observe during training) and out-of-distribution (**OOD**) environments

(associated with styles that we only observe during testing). Each dataset contains a set of annotated semantic categories, and we divide them into two sets: **normal** classes (content categories observed during training) and **novel** classes (content categories that should be distinguished from the normal ones during testing). For each sample, we have a style label and a novelty label (normal vs. novel).

**fMoW** comprises satellite images of various functional buildings. The style is defined by the location of the image, while the content is defined by the class of the observed building. To generate a greater shift between ID and OOD styles, we chose the environments to be given by the continent in which the images were taken. As such, we considered photos taken in Europe, America, Asia, and Africa to compose the ID styles, while those taken in Australia as OOD style. The content separation into normal and novel categories was randomly generated (see Appendix A.7.1).

**DomainNet** contains images of common objects in six different domains. The style is defined by the domain, while the content is defined by the object class. We separated the environments into ID: clipart, infograph, painting, and real and OOD: quickdraw and sketch. We randomly split the classes into normals and novelties (see Appendix A.7.2).

**COCOShift** is a synthetic dataset generated to allow an in-depth analysis of our approach. We combined segmented objects from COCO (Lin et al., 2014) with natural landscape imagery from Places365 (Zhou et al., 2017). The landscape images define the style of the data, while the object categories depict the content. We have grouped the landscape images into 9 categories (*e.g.* forest, mountain), each of an equal number of samples, and further split them into 5 ID and 4 OOD styles. The object categories were split into normal and novel classes by following a proper balancing between them. (see Appendix A.7.3). **Spuriousness**: For COCOShift, we deliberately introduced and varied the level of spurious correlations between style and content, similar to Kirichenko et al. (2023); Sagawa* et al. (2020). The spuriousness level ranges from 50% (balanced dataset) to 95% (where the normal class is strongly correlated with some environments, while we observe few samples in the rest of the environments). This results in the COCOShift benchmark, with 4 versions of spuriousness present in the training sets: COCOShift_balanced, COCOShift75, COCOShift90, and COCOShift95.

**Metrics**: For our ND experiments we report ROC-AUC metric, as the average performance over test environments. Unless otherwise specified, we report performance over OOD environments.

**Feature selection algorithms**: Besides our Stylist method, we consider InfoGain (Kraskov et al., 2004) and FisherScore (Gu et al., 2011), which we adapted to discard the environment-biased features. We denote those three methods as *'Env-Aware'* methods. As *'Not Env-Aware'* methods, we evaluate MAD (mean absolute difference), Dispersion (as the ratio between arithmetic and geometric mean), Variance, and PCA Loadings. We use all those methods to compute an individual score per feature (see Appendix A.1 for details).

**Novelty detection algorithms**: We observe the impact of our method on several ND solutions: OCSVM (Schölkopf et al., 1999), LOF (Breunig et al., 2000), and kNN (Angiulli & Pizzuti, 2002) with different variations (normalized or not at sample level, with 10 or 30 neighbors to measure variations). We also tested the impact in the state-of-the-art solution for OOD detection, kNN+ (Sun et al., 2022a), which trains a kNN on top of normalized samples, but on top of ResNet-18 features, fine-tuned using a supervised contrastive loss like in Khosla et al. (2020).

**Pretrained features**: We validate over multiple feature extractors, from different tasks, architectures, and datasets (supervised, multi-modal, contrastive, from basic ResNet to Visual Transformers, trained on ImageNet (Deng et al., 2009) and other larger datasets): ResNet-18, ResNet-34 (He et al., 2016), CLIP (Radford et al., 2021), ALIGN (Jia et al., 2021), BLIP-2 (Li et al., 2023b). Unless otherwise specified the experiments are conducted using ResNet-18 pretrained on ImageNet.

## 4.1 ROBUST NOVELTY DETECTION

**Stylist for Novelty Detection**   We evaluate how our selection affects the robustness of various Novelty Detection algorithms. Tab. 1 presents the initial performance using all features and the best result achieved by dropping environment-biased features as identified by *Stylist*. Notice how for almost all cases, using only a percentage of features improves the performance by up to 8%.

Table 1: Stylist feature ranking on ND Methods. Notice how, for almost all ND algorithms and dataset combinations, dropping top environment-biased features, as identified by Stylist, increases the ROC-AUC performance (see the improvement in green).

| | fMoW | | | DomainNet | | | COCOShift95 | | |
|---|---|---|---|---|---|---|---|---|---|
| | ROC-AUC ↑ | | % selected feat. | ROC-AUC ↑ | | % selected feat. | ROC-AUC ↑ | | % selected feat. |
| **ND Method** | all feat. | **Stylist** feat. | | all feat. | **Stylist** feat. | | all feat. | **Stylist** feat. | |
| OCSVM | 46.9 | 54.3 (+7.4) | 85 | 50.4 | 51.4 (+1.0) | 95 | 52.6 | 58.4 (+5.8) | 90 |
| LOF | 58.0 | 60.8 (+2.8) | 15 | 51.1 | 52.0 (+0.9) | 90 | 83.4 | 86.5 (+3.1) | 30 |
| kNN | 59.0 | 60.3 (+1.3) | 20 | 50.6 | 50.8 (+0.2) | 40 | 79.8 | 85.1 (+5.3) | 30 |
| kNN norm | 41.9 | 49.9 (+8.0) | 5 | 52.5 | 52.8 (+0.3) | 70 | 86.2 | 86.2 | 100 |
| kNN+ | 58.0 | 60.8 (+2.8) | 15 | 51.1 | 52.0 (+0.9) | 90 | 82.3 | 82.3 | 100 |

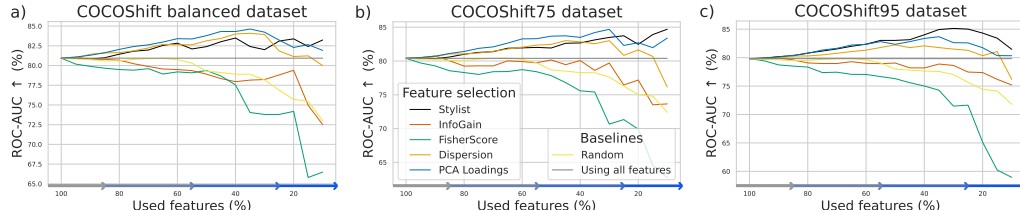

Figure 3: Feature selection algorithms. From left to right on the horizontal axis, we remove features according to the ranking of each feature selection algorithm. As the spuriousness level of the train set increases $(a) \rightarrow b) \rightarrow c)$), the performance of Stylist (in black) increases, while the performance of other methods decreases. This proves that our approach is better at identifying environment-biased features responsible for the spurious correlations. The reported ROC-AUC performance is for the same OOD sets in all three plots.

**Comparison with other feature selection algorithms** We compare in Fig. 3 between different methods of feature selection. For all algorithms, we drop the features ranked as the most irrelevant. We see that while we vary the spuriousness level in the training dataset, the relative order of the algorithms changes, showing that some perform better when working on a balanced dataset (like PCA based ones), while our *Stylist* works the best in difficult scenarios with an increased level of spurious correlations. See in Appendix A.1 the individual performances and notice in Appendix A.2 how those covariate shift results are consistent also for sub-population OOD shifts.

**Stylist robustness to dataset spuriousness level** To better understand the real cases, we further analyze the impact of spurious correlations in each step of our approach. We use datasets with various levels of *spuriousness* between style and content, in three setups (Fig. 4): **a)** use the same dataset in both algorithm steps; **b)** keep the spuriousity level fixed for Step2 while varying the spuriousity level for Step1 **c)** keep the spuriousity level fixed for Step1 while varying the spuriousity level for Step2. The dataset kept constant in b) and c) is COCOShift_balanced. We observe that having a higher degree of spuriousness in feature selection (Step 1), leads to better performance for our Stylist method. Nevertheless, in all cases, even in the most degenerated ones (with very high correlations to none), we see an increase after removing the top-ranked environment-biased features.

**Feature Selection vs. Dimensionality Reduction** Classical dimensionality reduction approaches (like PCA) address the idea of reducing space dimensionality while preserving or maximizing the most important information. In PCA, we can assume that a projection into the space of the principal components will produce a robust representation. Although this projection method is different from feature selection methods, since the first reprojects the features into a new space, instead of keeping certain existing features, we compare it with *Stylist*, in Fig. 5, for the robust novelty detection task. Consistently, for all datasets, *Stylist* selection performs better. We also combine *Stylist* with PCA, by applying an additional dimensionality reduction over the best percentage of features from *Stylist*. We observe an improvement in the curves, highlighting the potential of combining the two approaches.

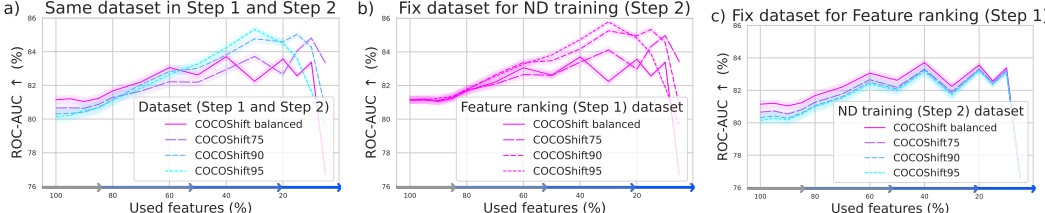

Figure 4: Dataset spuriousness impact. We vary the train set spuriousness level between style and content for the two steps of our algorithm. **a)** same dataset for both steps; **b)** fixed dataset (CO-COShift_balanced) for ND training in Step 2; **c)** fixed dataset (COCOShift_balanced) for Feature ranking in Step 1. Our method always manages to improve the ND performance (w.r.t. all features baseline), even in degenerated cases like 95% (or no) spurious correlation, in only one or in both steps (see the positive slopes in all curves).

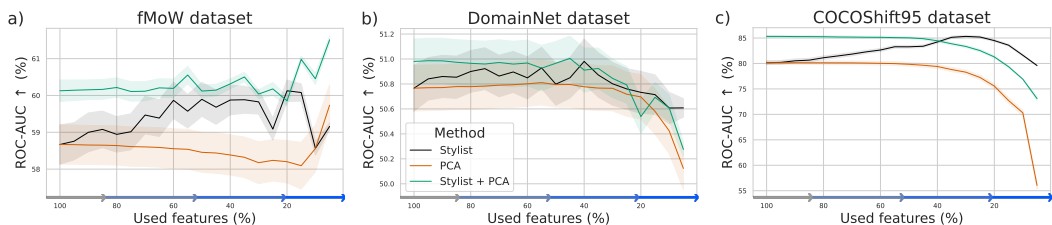

Figure 5: Features Selection vs Dimensionality Reduction. When comparing Stylist (black) with PCA (orange), we see that selection works better in all cases. Moreover, when combining the best selection percentage of Stylist with further dimensionality reduction using PCA (green), we observe an improvement (note that the green curve corresponds to different absolute numbers of features).

## 4.2 A GLIMPSE OF INTERPRETABILITY

Our approach ranks features based on their correlation with the style. To validate the quality of the ranking, we perform two experiments.

**a)** We build a synthetic scenario with disentangled features for style and content. We split each image from COCOShift_balanced train set into two images. One only contains the object (content), and the other only contains the background (style). We extract and combine features for the two images. The first half of the features are unrelated to style, while the second half is unrelated to content (for evaluation purposes we have considered that the first 50% of features are content features and the rest are style features). Further, we apply *Stylist* over this combined representation, on COCOShift_balanced dataset. In Fig. 6a) we present the results of our experiment. For the first 40% top-ranked environment-biased features, *Stylist* has a perfect accuracy score, with other *env-aware* methods (InfoGain and FisherScore) having also impressive scores of 99.1% and 99.7%, while the *non env-aware* methods performing significantly lower. In this scenario with disentangled features, env-aware methods consistently select as top features, those associated with the style.

**b)** In a real case scenario, where we know nothing about the meaning of each feature, we analyze the capacity of our top-scored environment-biased features to predict the style of an image. Starting from our COCOShift dataset, we build a balanced dataset, with no spurious correlations, for the task of classifying the style category of an image (1 out of 9). In Fig. 6b) we present the results of our experiment, where we have trained a classifier for each percent of features. We observe that with only a small fraction of the features, we achieve almost the maximum score for predicting the environment, showing that the top-ranked features are indeed predictive for the style. In contrast, when randomly selecting features, the same performance is achieved using significantly more features.

FisherScore selection method works very good in **a)**, which covers the perfect feature disentanglement case. But in the real scenario from **b)**, it fails below the random baseline. FisherScore looks for features that can effectively distinguish between different environments and also provide unique (or non-redundant) information compared to other features. It relies on computing full feature space

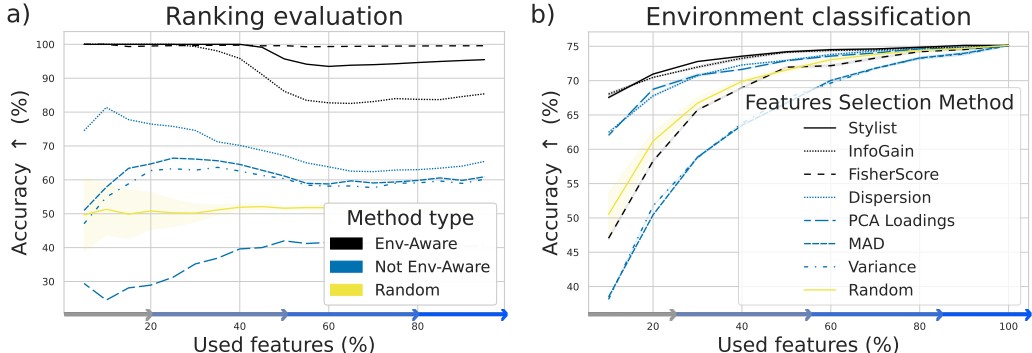

Figure 6: Features' focus analysis. **a)** In a controlled setup, with 50% of features being content-related and 50% being style-related, we evaluate how accurate is our Stylist ranking. We observe that the top-ranked 40% of environment-biased features are correctly identified with a perfect accuracy score. In fact, all *env-aware* methods have impressive results, overcoming *non env-aware* methods by a large margin. **b)** In a balanced setup, we have also evaluated the ability of our top-ranked environment-biased features to classify the style category of an image. Note that our approach reaches a high accuracy with only 5% of the top-ranked environment-biased features. This indicates that the identified features are indeed strongly correlated with the style.

Table 2: Feature extractors. Stylist improves the performance for all types of pretrained features considered, over all three datasets. For simplicity, we use only ResNet-18 in other experiments.

| Features | fMoW | | | DomainNet | | | COCOShift95 | | |
|---|---|---|---|---|---|---|---|---|---|
| | ROC-AUC ↑ | | % selected feat. | ROC-AUC ↑ | | % selected feat. | ROC-AUC ↑ | | % selected feat. |
| | all feat. | **Stylist** feat. | | all feat. | **Stylist** feat. | | all feat. | **Stylist** feat. | |
| ResNet-18 | 59.0 | 60.3 (+1.3) | 20 | 50.6 | 50.8 (+0.2) | 40 | 79.8 | 85.1 (+5.3) | 30 |
| ResNet-34 | 61.9 | 65.6 (+3.7) | 30 | 51.1 | 51.1 (+0.1) | 40 | 78.9 | 82.6 (+3.7) | 20 |
| CLIP | 54.3 | 55.5 (+1.3) | 25 | 60.8 | 61.5 (+0.8) | 30 | 94.5 | 94.9 (+0.4) | 95 |
| ALIGN | 54.6 | 56.2 (+1.6) | 40 | 60.6 | 60.8 (+0.3) | 75 | 89.6 | 89.7 (+0.1) | 80 |
| BLIP-2 | 58.6 | 59.1 (+0.4) | 15 | 65.1 | 65.8 (+0.7) | 20 | 96.7 | 96.8 (+0.1) | 95 |

distances when finding neighbours, but those distances are largely affected by the imperfect scenario, where features are intertwined and the feature extractor can contain an unbalanced ratio of style vs. content features. In contrast, *Stylist* analyzes distances between individual feature distributions, implicitly balancing the impact of content vs. style if let's say part of the spectrum looks similar because it represents the content part. This way, *Stylist* manages to be more robust in the real-case scenario like in **b)**.

### 4.3 ABLATIONS

**Feature extractors** We show in Tab. 2 that our feature selection method is model-agnostic, improving over 100% feature usage baseline, over a wide variety of pretrained models, coming from basic supervised classification, multi-modal and contrastive approaches.

**Stylist distance** We validate the algorithmic decisions of our proposed *Stylist* approach. To compute the per feature scores, we measure the per-feature distance in distribution, between any two training environments (Eq. 1), and combine those per-pair distances to obtain a more informative ranking, based on all training environments (Eq. 2). See in Fig. 7 how the per-pair ranking combinations do not influence the overall performance, while the used distance seems to be dataset-specific (symmetric KL is better on fMoW, while Wasserstein is better on DomainNet and the synthetic CO-COShift95). For simplicity, we have used Wasserstein distance with mean over the features per-pair scores in all our experiments. See Appendix A.3 for detailed scores.

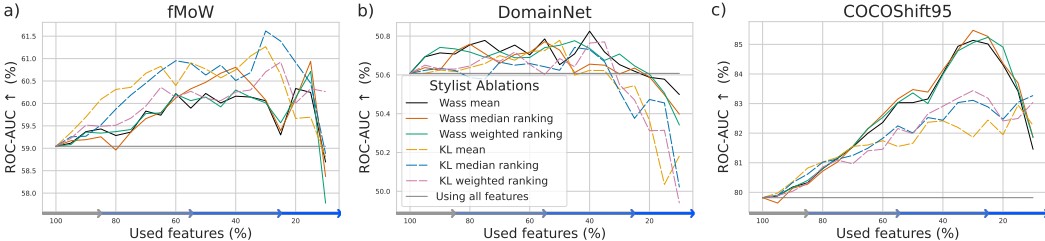

Figure 7: Stylist ablations. We vary both the distance metrics (based on Wasserstein or symmetric KL) and the ranking combination approaches for pairs of environments (mean, median ranking, weighted ranking). Notice how usually Wasserstein distances perform better (except for a) fMoW). Also, see how the ranking combination does not have a high influence on the result.

## 5 RELATED WORK

**OOD generalization**: Machine learning methods proved to have remarkable capabilities, but still being subject to mistakes when dealing with out-of-distribution data (Geirhos et al., 2020; Beery et al., 2018; Hendrycks et al., 2021; Lechner et al., 2022). **Invariant learning**: To tackle the changing distribution, one possible solution involves learning some invariant mechanisms of the data (Muandet et al., 2013; Peters et al., 2016; Arjovsky et al., 2019). IRM (Arjovsky et al., 2019) constraints the model such to obtain the same classifier in different environments, while vREx (Krueger et al., 2021) constrains the loss to have low variance across domains. The work of Ye et al. (2021) proves that features with small variations between training environments are important for out-of-distribution generalization. This also gives a formal motivation to our work. In Wang et al. (2022) a subspace of invariant features is determined through PCA of class-embeddings. A formalization of invariant learning is proposed in Wang & Veitch (2022) and suggests that depending on the structure of the data, different constraints should be used. **Domain adaptation**: Another popular approach is to employ test time adaptation (Wang et al., 2020; Gandelsman et al., 2022), but those methods require access to unlabeled samples of the target environment.

**Novelty detection**: Semantic anomaly detection (Ahmed & Courville, 2020) aims to detect only changes in some high-level semantic factors (*e.g.* object classes) as opposed to low-level cues (such as image artifacts). Methods like the ones in Tack et al. (2020); Sehwag et al. (2021); Winkens et al. (2020); Sun et al. (2022b) use a self-supervised method for anomaly or out-of-distribution detection while the methods in Li et al. (2021); Zhou et al. (2021); Reiss & Hoshen (2023) also adapt pre-trained extractors using contrastive methods. RedPanda (Cohen et al., 2023) method learns to ignore some irrelevant factors but achieves this using labels of such factors. Still, most works in this space only focus on settings containing only one type of factor, semantic or non-semantic, but not both. **Open Set Regonition** is a strongly related task, but it has access to semantic labels of the known classes. Open Set Recognition is typically approached in a supervised learning context, while Novelty Detection methods are often employed in an unsupervised context.

**Robust novelty detection**: We propose this term for the setting that contains both content and style factors, where the goal is to detect changes in content while being robust to style. This setting is introduced in Smeu et al. (2022) where they show that robustness methods based on multi-environment learning can help anomaly detection. Our work shows that a simple, but efficient method of ranking the feature invariance, improves the performance in the context of **robust novelty detection**.

## 6 CONCLUSIONS

In this work, we first propose *Stylist*, a feature selection method that searches for features focused more on the environment, which are irrelevant for a pursued task, by emphasizing the distribution distances between environments, at the feature level. Next, we prove that by dropping features for which our algorithm gives a high probability to be environment-biased, we improve the generalization performance of novelty detection, in the setup where both style and content distribution shifts. To validate our approach, we introduce COCOShift, a synthetic benchmark, on which we tested our solution on splits with various levels of spuriousness, alongside other two datasets, DomainNet and fMoW, composed of real sampled data.

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

Table 3: Feature selection methods.

| Selection Method | COCOShift_balanced | | | COCOShift75 | | | COCOShift95 | | |
|---|---|---|---|---|---|---|---|---|---|
| | ROC-AUC ↑ | | % selected feat. | ROC-AUC ↑ | | % selected feat. | ROC-AUC ↑ | | % selected feat. |
| | all feat. | **kept** feat. | | all feat. | **kept** feat. | | all feat. | **kept** feat. | |
| **Stylist (ours)** | 80.9 | 83.5 (+2.6) | 40 | 80.4 | **84.7** (+4.3) | 10 | 79.8 | **85.1** (+5.3) | 30 |
| InfoGain | 80.9 | 81.0 | 95 | 80.4 | 80.7 (+0.3) | 90 | 79.8 | 79.9 (+0.1) | 90 |
| FisherScore | 80.9 | 80.9 | 100 | 80.4 | 80.4 | 100 | 79.8 | 79.8 | 100 |
| MAD | 80.9 | 80.9 | 100 | 80.4 | 80.4 | 100 | 79.8 | 79.8 | 100 |
| Dispersion | 80.9 | 84.1 (+3.2) | 35 | 80.4 | 83.0 (+2.6) | 45 | 79.8 | 82.2 (+2.4) | 50 |
| Variance | 80.9 | 80.9 | 100 | 80.4 | 80.4 | 100 | 79.8 | 79.8 | 100 |
| PCA Loadings | 80.9 | **84.6** (+3.7) | 35 | 80.4 | **84.7** (+4.3) | 30 | 79.8 | 83.7 (+3.9) | 35 |

# A  APPENDIX

## A.1  FEATURE SELECTION METHODS - DETAILED

We show in Tab. 3 individual results for multiple feature selection algorithms, grouped into env-aware ones and algorithms that are not env-aware. Please note that we adapt basic algorithms for feature selection to make them env-aware.

Considered feature selection methods:

- env-aware
  - InfoGain (Kraskov et al., 2004): We adapt the method to the env-ware setup. We compute the mutual information between each feature and the style labels. High scores indicate a higher dependency between feature and style labels → environment-biased feature.
  - FisherScore (Gu et al., 2011): We adapt the method to the env-aware setup. We rank the features based on their relevance for the classification of style categories.
- non env-aware
  - MAD: For one feature, compute the average of absolute differences between each sample value and the mean value. High MAD values indicate a high discriminatory power.
  - Dispersion: This is computed as the ratio of arithmetic and geometric means. High dispersion implies a higher discriminatory power.
  - Variance: Generally you can use variance to discard zero variance features as being completely uninformative. We have ranked the features based on their variance, considering high-variance features as being more informative.
  - PCA Loadings: We compute the contribution of each feature to the set of principal components identified by PCA.

## A.2  DIFFERENT SHIFT ROBUSTNESS

We analyzed in Fig. 8 what is the impact of *Stylist* when we consider *Sub-Population shifts*, in extension to *Covariate shifts*, presented in the main paper. The testing dataset in this case is balanced, ID with the first plot (with no correlation between style and content). Our method manages to improve its performance when compared with other feature selectors, as the training and testing become more OOD. The observations are similar in both kinds of shifts.

## A.3  STYLIST ABLATION DISTANCES

In Tab. 4 we show individual scores when using symmetric KL or Wasserstein to measure per feature distribution distances between any two training environments. We combine the score or ranking

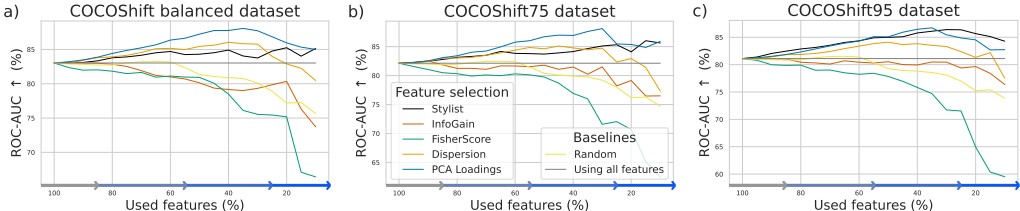

Figure 8: Feature selection algorithms. The reported ROC-AUC performance is for a testing dataset coming from the same (ID) distribution with a), showing that similar observations remains for sub-population OOD shifts.

Table 4: Ablation distance and ranking combining. We notice here that all variants of distances and feature ranking combinations manage to improve. Depending on the dataset, there are different chooses to make as hyper-parameters. We use for Stylist the mean of Wasserstein distances over all training pairs.

| Distance | Method | fMoW | | | DomainNet | | | COCOShift95 | | |
|---|---|---|---|---|---|---|---|---|---|---|
| | | ROC-AUC ↑ | | % selected feat. | ROC-AUC ↑ | | % selected feat. | ROC-AUC ↑ | | % selected feat. |
| | | all feat. | **Stylist** feat. | | all feat. | **Stylist** feat. | | all feat. | **Stylist** feat. | |
| **Wasserstein** | **mean** | 59.0 | 60.3 (+1.3) | 20 | 50.6 | 50.8 (+0.2) | 40 | 79.8 | 85.1 (+5.3) | 30 |
| | median | 59.0 | 60.6 (+1.5) | 15 | 50.6 | 50.8 (+0.2) | 75 | 79.8 | 84.2 (+4.4) | 20 |
| | median ranking | 59.0 | 60.9 (+1.9) | 15 | 50.6 | 50.8 (+0.2) | 55 | 79.8 | 85.5 (+5.7) | 30 |
| | weighted mean ranking | 59.0 | 60.7 (+1.7) | 15 | 50.6 | 50.8 (+0.2) | 45 | 79.8 | 85.2 (+5.4) | 25 |
| **KL symmetric** | mean | 59.0 | 61.3 (+2.2) | 30 | 50.6 | 50.8 (+0.2) | 50 | 79.8 | 83.0 (+3.1) | 15 |
| | median | 59.0 | 60.7 (+1.6) | 50 | 50.6 | 50.8 (+0.2) | 45 | 79.8 | 83.7 (+3.8) | 25 |
| | median ranking | 59.0 | 61.6 (+2.6) | 30 | 50.6 | 50.7 (+0.1) | 45 | 79.8 | 83.3 (+3.5) | 10 |
| | weighted mean ranking | 59.0 | 60.9 (+1.9) | 25 | 50.6 | 50.8 (+0.2) | 35 | 79.8 | 83.4 (+3.6) | 30 |

obtained, over all pairs of training envs, using mean, median, median ranking or weighted mean ranking.

## A.4 HYPERPARAMETERS ANALYSIS

Stylist is robust to the choice of hyperparameters. We emphasize the constant improvement illustrated below.

Considered hyperparameters:

- The **feature extractor**:
  - In Tabl 2 we provide an analysis of the performance w.r.t. the considered feature extractor. Our Stylist, improves the performance regardless of the selected feature extractor
- The **distance metric** and the **combination between pairwise distances** employed in Step1 of our algorithm:
  - In Figure 7 we provide an ablation regarding this matter. We highlight the performance improvement of our Stylist, irrespective of the considered configuration.
- The **percent of selected features** for Step2 of our algorithm:

- As highlighted in Figure3, Stylist consistently improves over the baseline w.r.t. the percent of considered features, proving that the provided feature ranking is relevant for the novelty detection problem. To select an optimal percent of features per setup, we employ a validation step, analyzing either the performance on an ID validation set or the performance on an OOD test set. As highlighted in the Table 5, there is a very small variance between those techniques ($< 0.07$ for ResNet-18 features, and even $0$ for CLIP features), highlighting that the performance is stable. This is an important property of our algorithm, managing to improve OOD performance, with ID chosen hyperparameters.

- The **Novelty Detection Algorithm**:
  - Table 1 shows the impact of our Stylist on different novelty detection algorithms. It showcases a consistent improvement over baselines in almost all scenarios.
  - For our default novelty detection algorithm, which is kNN, we provide in Table 5 an ablation regarding the choice of the number of neighbours (k). We highlight the robustness of the approach w.r.t. k, as the variance is in range $[0.002, 0.004]$ for CLIP and in range $[0.013, 0.160]$ for ResNet-18.

## A.5 ANALYSIS OF STYLIST WITH CLIP FEATURES

We provide additional experimental analysis in Table 6 and Table 7, considering CLIP as the feature extractor. We highlight the consistent improvement of our approach.

## A.6 FUTURE WORK

We leave here several unexplored directions, that we consider valuable for further investigations:

1. Analyze the impact of the pretrained feature extractor, looking after different axes of variation: supervised/unsupervised pretraining, high/low disentanglement. And going even further, find an unsupervised way to choose the best feature extractor, given a dataset.

2. Explore more complex algorithms for ranking, based on the same principle of emphasizing the intra and inter environment distances. More related to the algorithm, explore an unsupervised manner to choose the best percent of features to keep.

3. Take the approach beyond novelty detection, analyzing the performance improvement of feature ranking and selection w.r.t. other supervised approaches for OOD robustness.

Table 5: Hyperparameters analysis

| Features | Optimal nr of features selection approach | k | COCOShift_balanced ROC-AUC ↑ all feat. | Stylist feat. | % sel. feat. | COCOShift75 ROC-AUC ↑ all feat. | Stylist feat. | % sel. feat. | COCOShift95 ROC-AUC ↑ all feat. | Stylist feat. | % sel. feat. |
|---|---|---|---|---|---|---|---|---|---|---|---|
| CLIP | Based on ID val set | 5 | 94.96 | 95.29 (+0.33) | 95 | 94.77 | 95.11 (+0.34) | 95 | 94.59 | 94.97 (+0.38) | 95 |
| | | 10 | 95.07 | 95.38 (+0.31) | 95 | 94.88 | 95.22 (+0.34) | 95 | 94.65 | 95.02 (+0.37) | 95 |
| | | 20 | 95.08 | 95.41 (+0.33) | 95 | 94.91 | 95.25 (+0.34) | 95 | 94.61 | 94.99 (+0.38) | 95 |
| | | 30 | 95.06 | 95.40 (+0.34) | 95 | 94.88 | 95.21 (+0.33) | 95 | 94.53 | 94.92 (+0.39) | 95 |
| | Var. w.r.t. k | | 0.003 | 0.003 | - | 0.004 | 0.003 | - | 0.002 | 0.002 | - |
| | Based on OOD test set | 5 | 94.96 | 95.29 (+0.33) | 95 | 94.77 | 95.11 (+0.34) | 95 | 94.59 | 94.97 (+0.38) | 95 |
| | | 10 | 95.07 | 95.38 (+0.31) | 95 | 94.88 | 95.22 (+0.34) | 95 | 94.65 | 95.02 (+0.37) | 95 |
| | | 20 | 95.08 | 95.41 (+0.33) | 95 | 94.91 | 95.25 (+0.34) | 95 | 94.61 | 94.99 (+0.38) | 95 |
| | | 30 | 95.06 | 95.40 (+0.34) | 95 | 94.88 | 95.21 (+0.33) | 95 | 94.53 | 94.92 (+0.39) | 95 |
| | Var. w.r.t. k | | 0.003 | 0.003 | - | 0.004 | 0.003 | - | 0.002 | 0.002 | - |
| | Variance w.r.t. feat. sel | | - | 0 | - | - | 0 | - | - | 0 | - |
| ResNet-18 | Based on ID val set | 5 | 81.51 | 83.58 (+2.07) | 10 | 81.10 | 84.96 (+3.86) | 10 | 80.72 | 85.53 (+4.81) | 25 |
| | | 10 | 81.36 | 83.51 (+2.15) | 10 | 80.91 | 84.90 (+3.99) | 10 | 80.48 | 85.38 (+4.90) | 25 |
| | | 20 | 81.12 | 83.36 (+2.24) | 10 | 80.62 | 84.79 (+4.17) | 10 | 80.11 | 85.16 (+5.05) | 25 |
| | | 30 | 80.93 | 83.24 (+2.31) | 10 | 80.40 | 84.71 (+4.31) | 10 | 79.82 | 85.01 (+5.19) | 25 |
| | Var. w.r.t. k | | 0.065 | 0.023 | - | 0.096 | 0.013 | - | 0.160 | 0.052 | - |
| | Based on OOD test set | 5 | 81.51 | 84.08 (+2.57) | 40 | 81.10 | 84.96 (+3.86) | 10 | 80.72 | 85.66 (+4.94) | 30 |
| | | 10 | 81.36 | 83.92 (+2.56) | 40 | 80.91 | 84.90 (+3.99) | 10 | 80.48 | 85.51 (+5.03) | 30 |
| | | 20 | 81.12 | 83.68 (+2.56) | 40 | 80.62 | 84.79 (+4.17) | 10 | 80.11 | 85.29 (+5.18) | 30 |
| | | 30 | 80.93 | 83.49 (+2.56) | 40 | 80.40 | 84.71 (+4.31) | 10 | 79.82 | 85.13 (+5.31) | 30 |
| | Var. w.r.t. k | | 0.065 | 0.069 | - | 0.096 | 0.013 | - | 0.160 | 0.054 | - |
| | Variance w.r.t. feat. sel | | - | 0.068 | - | - | 0 | - | - | 0.008 | - |

Table 6: Stylist feature ranking on ND Methods - CLIP features

| Features | Optimal nr of features selection approach | ND Algo. | fMoW | | | DomainNet | | | COCOShift95 | | |
|---|---|---|---|---|---|---|---|---|---|---|---|
| | | | ROC-AUC ↑ | | % sel. feat. | ROC-AUC ↑ | | % sel. feat. | ROC-AUC ↑ | | % sel. feat. |
| | | | all feat. | Stylist feat. | | all feat. | Stylist feat. | | all feat. | Stylist feat. | |
| CLIP | Based on ID val set | OCSVM | 51.97 | 51.97 | 100 | 53.71 | 55.98 (+2.27) | 85 | 44.26 | 64.00 (+19.74) | 10 |
| | | LOF | 55.57 | 57.30 (+1.73) | 30 | 58.62 | 59.45 (+0.83) | 50 | 91.19 | 91.34 (+0.15) | 95 |
| | | kNN | 54.26 | 54.47 (+0.21) | 90 | 60.78 | 61.20 (+0.42) | 50 | 94.53 | 94.92 (+0.39) | 95 |
| | | kNN norm | 48.04 | 47.39 (-0.65) | 70 | 61.42 | 61.76 (+0.34) | 85 | 91.23 | 92.67 (+1.44) | 50 |
| | Based on OOD test set | OCSVM | 51.97 | 52.76 (+0.79) | 80 | 53.71 | 55.98 (+2.27) | 85 | 44.26 | 64.00 (+19.74) | 10 |
| | | LOF | 55.57 | 57.30 (+1.73) | 30 | 58.62 | 59.71 (+1.09) | 80 | 91.19 | 91.34 (+0.15) | 95 |
| | | kNN | 54.26 | 55.54 (+1.28) | 25 | 60.78 | 61.54 (+0.76) | 30 | 94.53 | 94.92 (+0.39) | 95 |
| | | kNN norm | 48.04 | 50.03 (+1.99) | 10 | 61.42 | 62.03 (+0.61) | 40 | 91.23 | 92.90 (+1.67) | 70 |
| | Variance w.r.t. feat. sel | | - | 1.092 | - | - | 0.032 | - | - | 0.007 | - |

Table 7: Comparison to feature selection methods - CLIP features

| Features | Optimal nr of features selection approach | Selection Algo. | fMoW | | | DomainNet | | | COCOShift95 | | |
|---|---|---|---|---|---|---|---|---|---|---|---|
| | | | ROC-AUC ↑ | | % sel. feat. | ROC-AUC ↑ | | % sel. feat. | ROC-AUC ↑ | | % sel. feat. |
| | | | all feat. | Stylist feat. | | all feat. | Stylist feat. | | all feat. | Stylist feat. | |
| CLIP | Based on ID val set | Stylist | 54.26 | 54.47 (+0.21) | 90 | 60.78 | 61.20 (+0.42) | 50 | 94.53 | 94.92 (+0.39) | 95 |
| | | InfoGain | 54.26 | 54.44 (+0.18) | 90 | 60.78 | 61.21 (+0.43) | 45 | 94.53 | 94.53 | 100 |
| | | FisherScore | 54.26 | 54.39 (+0.13) | 60 | 60.78 | 60.83 (+0.05) | 95 | 94.53 | 94.53 | 100 |
| | | MAD | 54.26 | 54.26 | 100 | 60.78 | 60.78 | 100 | 94.53 | 94.79 (+0.26) | 95 |
| | | Dispersion | 54.26 | 54.26 | 100 | 60.78 | 60.82 (+0.04) | 80 | 94.53 | 94.81 (+0.28) | 95 |
| | | Variance | 54.26 | 54.26 | 100 | 60.78 | 60.78 | 100 | 94.53 | 94.76 (+0.23) | 95 |
| | | PCA loadings | 54.26 | 54.61 (+0.35) | 80 | 60.78 | 61.25 (+0.47) | 75 | 94.53 | 94.66 (+0.13) | 85 |
| | Based on OOD test set | Stylist | 54.26 | 55.54 (+1.28) | 25 | 60.78 | 61.54 (+0.76) | 30 | 94.53 | 94.92 (+0.39) | 95 |
| | | InfoGain | 54.26 | 56.84 (+2.58) | 5 | 60.78 | 61.53 (+0.75) | 30 | 94.53 | 94.53 | 100 |
| | | FisherScore | 54.26 | 54.55 (+0.29) | 25 | 60.78 | 60.83 (+0.05) | 95 | 94.53 | 94.53 | 100 |
| | | MAD | 54.26 | 55.19 (+0.93) | 5 | 60.78 | 60.78 | 100 | 94.53 | 94.79 (+0.26) | 95 |
| | | Dispersion | 54.26 | 54.26 | 100 | 60.78 | 60.92 (+0.14) | 60 | 94.53 | 94.81 (+0.28) | 95 |
| | | Variance | 54.26 | 54.26 | 100 | 60.78 | 60.78 | 100 | 94.53 | 94.76 (+0.23) | 95 |
| | | PCA loadings | 54.26 | 57.52 (+3.26) | 10 | 60.78 | 61.33 (+0.55) | 55 | 94.53 | 94.76 (+0.23) | 95 |

## A.7 BENCHMARKS

### A.7.1 FMOW

- Content (functional purpose of buildings):
  - **normal**: airport, airport terminal, barn, burial site, car dealership, dam, debris or rubble, educational institution, electric substation, fountain, gas station, golf course, hospital, interchange, multi-unit residential, parking lot or garage, police station, port, railway bridge, recreational facility, road bridge, runway, shipyard, shopping mall, solar farm, space facility, surface mine, swimming pool, waste disposal, water treatment facility, zoo
  - **novel**: airport hangar, amusement park, aquaculture, archaeological site, border checkpoint, construction site, crop field, factory or powerplant, fire station, flooded road, ground transportation station, helipad, impoverished settlement, lake or pond, lighthouse, military facility, nuclear powerplant, office building, oil or gas facility, park, place of worship, prison, race track, single-unit residential, smokestack, stadium, storage tank, toll booth, tower, tunnel opening, wind farm
- Style (geographical area):
  - **ID**: Europe, America, Asia, Africa
  - **OOD**: Australia

- Number of samples:
  - OOD test set: 3469
  - ID train set: 55288
  - ID test set: 13817
  - ID val set: 6911

### A.7.2 DOMAINNET

- Content (object classes):
  - **normal**: aircraft carrier, angel, animal migration, apple, arm, backpack, barn, basketball, bed, belt, birthday cake, blackberry, blueberry, book, boomerang, bowtie, brain, bread, bucket, butterfly, cactus, cake, camouflage, cannon, carrot, cat, cello, chandelier, circle, cloud, coffee cup, computer, cookie, couch, crab, crayon, crocodile, cruise ship, diamond, diving board, dog, donut, door, dresser, drill, drums, duck, ear, elbow, envelope, eraser, fan, fence, flower, flying saucer, fork, frog, garden, guitar, hand, headphones, helicopter, helmet, hexagon, hockey stick, horse, hospital, hot air balloon, hot dog, hourglass, house, hurricane, jacket, jail, kangaroo, knife, laptop, leg, light bulb, lighter, lightning, lipstick, lobster, map, microphone, microwave, mountain, moustache, mug, mushroom, necklace, nose, owl, paint can, paintbrush, palm tree, parachute, parrot, peanut, pear, peas, piano, pig, pillow, pineapple, pizza, police car, pond, postcard, power outlet, radio, rain, rake, remote control, roller coaster, sailboat, saw, saxophone, screwdriver, sea turtle, see saw, shark, shorts, skull, sleeping bag, snail, snowman, soccer ball, spider, spoon, square, stairs, star, stethoscope, stitches, stop sign, strawberry, streetlight, string bean, submarine, suitcase, sun, swan, sweater, swing set, syringe, table, teapot, teddy-bear, telephone, television, tennis racquet, tent, toaster, toe, tooth, traffic light, train, tree, trombone, truck, trumpet, umbrella, underwear, van, vase, violin, whale, wheel, wine bottle, wristwatch, zebra
  - **novel**: airplane, alarm clock, ambulance, ant, anvil, asparagus, axe, banana, bandage, baseball, baseball bat, basket, bat, bathtub, beach, bear, beard, bee, bench, bicycle, binoculars, bird, bottlecap, bracelet, bridge, broccoli, broom, bulldozer, bus, bush, calculator, calendar, camel, camera, campfire, candle, canoe, car, castle, ceiling fan, cell phone, chair, church, clarinet, clock, compass, cooler, cow, crown, cup, dishwasher, dolphin, dragon, dumbbell, elephant, eye, eyeglasses, face, feather, finger, fire hydrant, fireplace, firetruck, fish, flamingo, flashlight, flip flops, floor lamp, foot, frying pan, garden hose, giraffe, goatee, golf club, grapes, grass, hamburger, hammer, harp, hat, hedgehog, hockey puck, hot tub, house plant, ice cream, key, keyboard, knee, ladder,

lantern, leaf, lighthouse, line, lion, lollipop, mailbox, marker, matches, megaphone, mermaid, monkey, moon, mosquito, motorbike, mouse, mouth, nail, ocean, octagon, octopus, onion, oven, panda, pants, paper clip, passport, pencil, penguin, pickup truck, picture frame, pliers, pool, popsicle, potato, purse, rabbit, raccoon, rainbow, rhinoceros, rifle, river, rollerskates, sandwich, school bus, scissors, scorpion, sheep, shoe, shovel, sink, skateboard, skyscraper, smiley face, snake, snorkel, snowflake, sock, speedboat, spreadsheet, squiggle, squirrel, steak, stereo, stove, sword, t-shirt, The Eiffel Tower, The Great Wall of China, The Mona Lisa, tiger, toilet, toothbrush, toothpaste, tornado, tractor, triangle, washing machine, watermelon, waterslide, windmill, wine glass, yoga, zigzag

- Style (manner of depiction):
  - **ID**: real, painting, clipart, infograph
  - **OOD**: sketch, quickdraw

- Number of samples:
  - OOD test set: 242886
  - ID train set: 142026
  - ID test set: 35313
  - ID val set: 17753

### A.7.3 COCOSHIFT

Each COCOShift environment is composed of 5 closely related categories of Places365 as follows:

- **forest**: forest, rainforest, bamboo_forest, forest_path, forest_road,
- **mountain**: mountain, mountain_snowy, glacier, mountain_path, crevasse
- **seaside**: beach, coast, ocean, boathouse, beach_house"
- **garden**: botanical_garden, formal_garden, japanese_garden, vegetable_garden, greenhouse
- **field**: field_cultivated, field_wild, wheat_field, corn_field, field_road
- **rock**: badlands, butte, canyon, cliff, grotto
- **lake**: lake, lagoon, swamp, marsh, hot_spring
- **farm**: orchard, vineyard, farm, rice_paddy, pasture
- **sport_field**: soccer_field, football_field, golf_course, baseball_field, athletic_field

On the other hand, we worked with superclasses from COCO (Lin et al., 2014) such that there would be a significant shift between classes.

To make sure that the content is identifiable from each image (as many COCO segmentations provide little content without it's context), we tested each generated image against CLIP (Radford et al., 2021). Specifically, we took a given merged picture into COCOShift dataset if CLIP could correctly identify the content between the selected COCO classes (not superclasses) listed below. The same test is effectuated on images of segmentations over white backgrounds.

- Content (object category):
  - **normal**: food (composed of classes: hot dog, cake, donut, carrot, sandwich, broccoli, banana, apple, pizza, orange)
  - **novel**: electronic (composed of classes: remote, laptop, tv, cell phone, keyboard), kitchen (composed of classes: bottle, cup, wine glass, knife, fork, bowl, spoon)
- Style (background area surrounding the object):
  - **ID**: forest, mountain, field, rock, farm
  - **OOD**: lake, seaside, garden, sport field

- Number of samples:

- – OOD test set: 13013
- – ID test set: 1623
- – COCOShift_balanced
  - * ID train set: 7033
  - * ID val set: 880
- – COCOShift75
  - * ID train set: 4221
  - * ID val set: 529
- – COCOShift90
  - * ID train set: 3284
  - * ID val set: 412
- – COCOShift95
  - * ID train set: 3037
  - * ID val set: 379

- **Spurious correlation** sets are variants of train sets used for the synthetic COCOShift dataset, where we eliminate samples to create spuriousity.
  - – **COCOShift_balanced**: normal samples are uniformly distributed among the ID environments ($\approx 1.4k$ samples per environment)
  - – **COCOShift75**: environments [farm, mountain] have $\approx 1.4k$ normal samples, while environments [rock, forest, field] have $\approx 400$ normal samples
  - – **COCOShift90**: environments [farm, mountain] have $\approx 1.4k$ normal samples, while environments [rock, forest, field] have $\approx 150$ normal samples
  - – **COCOShift95**: environments [farm, mountain] have $\approx 1.4k$ normal samples, while environments [rock, forest, field] have $\approx 70$ normal samples

