# OpenReview forum: "Stylist: Style-Driven Feature Ranking for Robust Novelty Detection"
_ICLR.cc/2024/Conference — Submitted to ICLR 2024_

### Official Review · Reviewer_6JHA · 2023-10-25

**Soundness:** 2 fair
**Presentation:** 2 fair
**Contribution:** 2 fair
**Rating:** 5
**Confidence:** 4

**Summary:**

Novelty detection is a nuanced problem, as one cannot distinguish between novel styles to be generalized and novel contents to be filtered. To distinguish these two cases, this paper considers a multi-environment setup in which the user is aware of multiple datasets with the same content but different styles. Given a pre-trained backbone, the paper introduces a feature selection method called Stylist, which selects the more env-invariant features by computing perturbations across environments. By focusing solely on the env-invariant features, Stylist improves upon previous feature-based novelty detection methods.

**Strengths:**

Distinguishing novel styles and contents is an important problem in the literature on uncertainty and robustness. Prior works have mostly focused on a single task, either generalizing to novel styles (i.e., domain generalization) or filtering novel contents (i.e., novelty detection). Bridging the gap between both fields is an important research direction.

**Weaknesses:**

**Limited scope**

The paper addresses a minor tweak of an existing problem. In doing so, it proposes an intuitive method that can be integrated with existing techniques, resulting in a clear improvement over the baseline.
While tackling a niche problem can be an easy way to write a paper, I believe that in representative venues like ICLR, the focus should be on addressing the core of the problem rather than opting for low-hanging fruit.

The authors could explore more impactful problems. For example, a single scalar uncertainty metric fails to differentiate between novel styles and contents. The proposed Stylist can be viewed as introducing a two-dimensional uncertainty to address this limitation.
Building on this concept, the authors might consider developing a unified framework that addresses both domain generalization and novelty detection in real-world situations.

---
**Comparison with invariant learning**

The paper aims to select domain-invariant features in a post-hoc manner. However, features can also be trained to be invariant using methods such as IRM.
A comparison between the proposed post-hoc feature selection and invariant learning is necessary, considering both domain generalization (tables in IRM) and novelty detection (tables in this paper).

---
**CLIP should be the default backbone choice**

In the methods section, the paper states, "our approach leverages pretrained embeddings with extensive coverage across various content and style categories."
However, the paper uses ResNet-18 pretrained on ImageNet for the main results, which does not align with the statement.

The paper should primarily consider models like CLIP, which are trained on diverse datasets and known to be robust to domain shifts.
The paper presents the CLIP results in Table 2, and the performance gap is significantly lower than that of the (domain-variant) ResNet.
Thus, the overall benefits of the proposed method may be exaggerated, and it would be more appropriate to conduct the main results using the less domain-variant models like CLIP.

---
**Hyperparameter selection**

The OOD detection competitors considered in the paper apply simple methods, such as OCSVM or kNN, on top of a fixed representation.
Given that this paper refines the representation, it is unsurprising that the proposed method enhances the original representation when appropriate hyperparameters are selected.

Furthermore, how were the hyperparameters selected? It seems they were chosen for optimal performance in the reported table.
However, (1) hyperparameters should not be spoiled by the test cases, and (2) the OOD detection method should be robust to unseen samples.
The correct approach is to select hyperparameters in a validation domain and evaluate their performance on novel test domains.

---
**Dataset contribution is marginal**

(1) The necessity of COCOShift95 over prior works is unclear.\
The paper introduces a new benchmark called COCOShift95, where objects are cut and shifted to different backgrounds.
Consequently, this benchmark is artificial and does not demand substantial effort, unlike previous works such as fMoW and DomainNet, which require considerable effort to collect realistic images.
What new insights can be gained from COCOShift95, aside from adding yet another column to the table?

(2) The cut and paste strategy has been considered in prior works.\
The approach of creating an artificial dataset through cut and paste has been proposed in previous works, such as Waterbirds and Background Challenge.
Additionally, why not use more natural benchmarks, such as MetaShift, which also include multi-domain images from COCO?
Therefore, the technical contribution of the proposed dataset collection strategy is also not convincing.

---
**Nitpicks**

- "+0.0" should not be highlighted in green in Table 3. It exaggerates the benefit of the proposed method.
- "Stylist" seems to be too ambiguous as a name for the feature selection method in multi-domain novelty detection.

**Questions:**

1. Why posthoc feature selection instead of using an invariant backbone like IRM or CLIP?
2. Were the hyperparameters chosen from the val set? Are they generalizable to unseen domains?

---

> ### Author Response · Authors · 2023-11-21
> **Response to Reviewer 6JHA - part 1**
>
> Thank you for your detailed comments and helpful insights. We have addressed the raised points in the comments below.
>
> **W1. Limited scope**
> Respectfully, we don’t consider the Stylist approach or its setup a minor tweak or a low-hanging fruit, but on the contrary, it could help address really impactful problems.
>
> First, If the reviewer referred to the downstream task impact:
>
> ***a) Robust Novelty Detection (ND) task***: Going from ND towards Robust ND does not create a niche, but quite the contrary, extending the solutions that work only in isolated/well-controlled lab scenarios (where the style distribution remains stable at test time) towards real-life scenarios ( where this is almost never the case). Our solution is particularly suitable for this task because it allows us to define what a novelty is, in an implicit way, by excluding what it is not (the style), in a very generic way and weakly supervised manner. As opposed to this, for instance, a supervised downstream task would focus on defining the content, rather than defining the non-content (the style). Such an approach requires lots of supervision and focus on covering all the variations for the content aspects.
>
> Secondly, if the reviewer referred to the main contribution impact, there is a large applicability to the idea:
>
> ***b) Biases***: Using Stylist, one can identify biases in a dataset, given a pre-trained representation by interpreting the environment-biased features. This could lead to better datasets in the field, and better understanding and discovery of the existing biases from commonly used data.
>
> ***c) Implicit Robustness***: Using Stylist, one can define the “space” for robustness in an implicit way, through environments (rather than doing it explicitly, with augmentations for instance). This leads to far more general and robust algorithms.
>
> For both b) and c) scenarios, you need a way to define (label) or find (maybe in an unsupervised way) intrinsic environments from the dataset.
>
> **W2 and Q1. Comparison with invariant learning**
>
> ***[Comparison with invariant learning]*** This is a very good point raised by the reviewer. There is a very important difference in the setup between the existing methods for learning features that are invariant to the environment (like IRM) and our Stylist:
>
> ***a)*** For computing the invariant features, all the others need both semantic (content) labels (namely content classes) and environment labels, while our Stylist only needs the environment labels. We argue that it is a huge effort to have those semantic labels, similar to the supervised vs unsupervised differences.
>
> ***b)*** Indeed, to continue the argument, one may say that we use pre-trained models. And this means we also need labels at some point. So, to balance the need for semantic labels, we can make the IRM also use pre-trained models this way: one can learn IRM to predict labels that a pre-trained method gives for the current dataset. But this is a tricky step, because it might result in a lot of knowledge loss, since for training the IRM, you will go beyond the pre-trained features, to a final target class that is not necessarily mapped over the new dataset. This way, the final supervision will greatly influence the representations, especially if there is not a 100% alignment between your new dataset and the pre-trained model training dataset (and this is a common case).
>
> ***c)*** Nevertheless, since semantic labels are available in our used datasets, we did an experiment that focuses on the complementarity between the methods, not on their differences (we explained before why this direct comparison is not fair): We first trained IRM on those semantic labels. Next, we applied Stylist on top of them. Interestingly, we see a 1% increase over using 100% of the features, showing that the two methods are somehow complementary.
>
> ***IRM*** - Invariant Risk Minimization: Martin Arjovsky, Léon Bottou, Ishaan Gulrajani, David Lopez-Paz

---

> > ### Author Response · Authors · 2023-11-21
> > **Response to Reviewer 6JHA - part 2**
> >
> > **W3. CLIP should be the default backbone choice**
> >
> > We agree that for the particular datasets considered in the current paper, CLIP is a more reliable representation, but for more general scenarios (see the use case below), ResNet-18 features may showcase a more realistic behavior. For a general problem, it may be difficult to collect a large and diverse dataset like the one employed for training CLIP, but rather be restricted to learn from a far, less representative set like in the case of ResNet-18.
> >
> > ***[use case]*** Camera algorithms that work on self-driving cars face lots of challenges regarding generalization properties. They need to be robust to various landscapes, building architectures, even car appearances, or other smaller particularities that can arise from one place to another. In a real-life scenario like this, getting a) semantic labels for samples is usually expensive. Also, in this case, b) environment labels are relatively cheap to get (e.g. group samples based on the location/city/region/country). Another important aspect, is the c) existence of good pre-trained representation (that generalizes well) for your specific task. This is a good real-world use case example in which our Stylist can take advantage of a) and b) and improve c) the quality of the representation, by identifying some spurious features.
> >
> > ***[CLIP experimental analysis]*** For the sake of completeness, we provide additional experimental analysis in the tables below, considering CLIP as the feature extractor. The provided analysis has been included in our manuscript. We highlight the consistent improvement of our approach.
> >
> > ***Stylist feature ranking on ND Methods***
> >
> > |Dataset | Features | Optimal nr. of features selection approach | ND Method | ROC-AUC all feat. | ROC-AUC Stylist feat. | % selected feat. |
> > |--|--|--|--|--|--|--|
> > |fMoW | CLIP | Based on ID val set | OCSVM | 51.97 | 51.97 | 100  |
> > | | | | LOF | 55.57 | 57.30 (+1.73) | 30 |
> > | | | | kNN | 54.26 | 54.47 (+0.21) | 90 |
> > | | | | kNN norm | 48.04 | 47.39 (-0.65) | 70 |
> > | | | Based on OOD test set | OCSVM | 51.97 | 52.76 (+0.79) | 80 |
> > | | | | LOF | 55.57 | 57.30 (+1.73) | 30 |
> > | | | | kNN | 54.26 | 55.54 (+1.28) | 25 |
> > | | | | kNN norm | 48.04 | 50.03 (+1.99) | 10 |
> > | | | **Variance w.r.t. features selection approach** | - | - | **1.092** | - |
> > |DomainNet | CLIP | Based on ID val set | OCSVM | 53.71 | 55.98 (+2.27) | 85 |
> > | | | | LOF | 58.62 | 59.45 (+0.83) | 50 |
> > | | | | kNN | 60.78 | 61.20 (+0.42) | 50 |
> > | | | | kNN norm | 61.42 | 61.76 (+0.34) | 85 |
> > | | | Based on OOD test set | OCSVM | 53.71 | 55.98 (+2.27) | 85 |
> > | | | | LOF | 58.62 | 59.71 (+1.09) | 80 |
> > | | | | kNN | 60.78 | 61.54 (+0.76) | 30 |
> > | | | | kNN norm | 61.42 | 62.03 (+0.61) | 40 |
> > | | | **Variance w.r.t. features selection approach** | - | - | **0.032** | - |
> > |COCOShift95 | CLIP | Based on ID val set | OCSVM | 44.26 | 64.00 (+19.74) | 10 |
> > | | | | LOF | 91.19 | 91.34 (+0.15) | 95 |
> > | | | | kNN | 94.53 | 94.92 (+0.39) | 95 |
> > | | | | kNN norm | 91.23 | 92.67 (+1.44) | 50 |
> > | | | Based on OOD test set | OCSVM | 44.26 | 64.00 (+19.74) | 10 |
> > | | | | LOF | 91.19 | 91.34 (+0.15) | 95 |
> > | | | | kNN | 94.53 | 94.92 (+0.39) | 95 |
> > | | | | kNN norm | 91.23 | 92.90 (+1.67) | 70 |
> > | | | **Variance w.r.t. features selection approach** | - | - | **0.007** | - |

---

> > > ### Author Response · Authors · 2023-11-21
> > > **Response to Reviewer 6JHA - part 3**
> > >
> > > ***Comparison between Stylist and different features selection methods***
> > >
> > > |Dataset | Features | Optimal nr. of features selection approach | Selection Method | ROC-AUC all feat. | ROC-AUC Stylist feat. | % selected feat. |
> > > |--|--|--|--|--|--|--|
> > > |fMoW | CLIP | Based on ID val set | **Stylist (ours)** | 54.26 | 54.47 (+0.21) |  90 |
> > > | | | | InfoGain | 54.26 | 54.44 (+0.18) | 90 |
> > > | | | | FisherScore | 54.26 | 54.39 (+0.13) | 60 |
> > > | | | | MAD | 54.26 | 54.26 | 100 |
> > > | | | | Dispersion | 54.26 | 54.26 | 100 |
> > > | | | | Variance | 54.26 | 54.26 | 100 |
> > > | | | | PCA loadings | 54.26 | 54.61 (+0.35) | 80 |
> > > | | | Based on OOD test set | **Stylist (ours)** | 54.26 | 55.54 (+1.28) |  25 |
> > > | | | | InfoGain | 54.26 | 56.84 (+2.58) | 5 |
> > > | | | | FisherScore | 54.26 | 54.55 (+0.29) | 25 |
> > > | | | | MAD | 54.26 | 55.19 (+0.93) | 5 |
> > > | | | | Dispersion | 54.26 | 54.26 | 100 |
> > > | | | | Variance | 54.26 | 54.26 | 100 |
> > > | | | | PCA loadings | 54.26 | 57.52 (+3.26) | 10 |
> > > |DomainNet | CLIP | Based on ID val set | **Stylist (ours)** | 60.78 | 61.20 (+0.42) | 50  |
> > > | | | | InfoGain | 60.78 | 61.21 (+0.43) | 45 |
> > > | | | | FisherScore | 60.78 | 60.83 (+0.05) | 95 |
> > > | | | | MAD | 60.78 | 60.78 | 100 |
> > > | | | | Dispersion | 60.78 | 60.82 (+0.04) | 80 |
> > > | | | | Variance | 60.78 | 60.78 | 100 |
> > > | | | | PCA loadings | 60.78 | 61.25 (+0.47) | 75 |
> > > | |  | Based on OOD test set | **Stylist (ours)** | 60.78 | 61.54 (+0.76) | 30  |
> > > | | | | InfoGain | 60.78 | 61.53 (+0.75) | 30 |
> > > | | | | FisherScore | 60.78 | 60.83 (+0.05) | 95 |
> > > | | | | MAD | 60.78 | 60.78 |  100 |
> > > | | | | Dispersion | 60.78 | 60.92 (+0.14) | 60 |
> > > | | | | Variance | 60.78 | 60.78 | 100 |
> > > | | | | PCA loadings | 60.78  | 61.33 (+0.55) | 55 |
> > > |COCOShift95 | CLIP | Based on ID val set | **Stylist (ours)** | 94.53 | 94.92 (+0.39) | 95  |
> > > | | | | InfoGain |94.53  | 94.53 | 100 |
> > > | | | | FisherScore |94.53  | 94.53 | 100 |
> > > | | | | MAD | 94.53 | 94.79 (+0.26) | 95 |
> > > | | | | Dispersion | 94.53 | 94.81 (+0.28) | 95 |
> > > | | | | Variance |94.53  | 94.76 (+0.23) | 95 |
> > > | | | | PCA loadings | 94.53 | 94.66 (+0.13) | 85 |
> > > | |  | Based on OOD test set | **Stylist (ours)** | 94.53 | 94.92 (+0.39) | 95  |
> > > | | | | InfoGain |94.53  | 94.53 | 100 |
> > > | | | | FisherScore |94.53  | 94.53 | 100 |
> > > | | | | MAD | 94.53 | 94.79 (+0.26) | 95 |
> > > | | | | Dispersion | 94.53 | 94.81 (+0.28) | 95 |
> > > | | | | Variance |94.53  | 94.76 (+0.23) | 95 |
> > > | | | | PCA loadings | 94.53 | 94.76 (+0.23) | 95 |
> > >
> > >
> > >
> > > **W4 and Q2. Hyperparameter selection**
> > >
> > > ***[hyperparameters]*** Stylist is robust to the choice of hyperparameters. We emphasize the constant improvement illustrated below.
> > >
> > > Hyperparameters:
> > >
> > > * ***The feature extractor:***
> > > In Table 2 of our manuscript we provide an analysis of the performance w.r.t. the considered feature extractor. Our Stylist, improves the performance regardless of the selected feature extractor.
> > >
> > > * ***The distance metric*** and the ***combination between pairwise distances*** employed in Step1 of our algorithm:
> > > In Figure 7 of our manuscript we provide an ablation study regarding this matter. We highlight the performance improvement of our Stylist, irrespective of the considered configuration.
> > >
> > > * ***The percent of selected features***, for Step2 of our algorithm:
> > > As highlighted in Figure 3, Stylist consistently improves over the baseline w.r.t. the percent of considered features, proving that the provided feature ranking is relevant for the novelty detection problem. To select an optimal percent of features per setup, we employ a validation step, analyzing either the performance on an ID validation set or the performance on an OOD test set. As highlighted in the table below, there is a very small performance variance between those techniques (< 0.07 for ResNet-18 features, and even 0 for CLIP features), highlighting that the performance is stable. This is an important property of our algorithm, managing to improve OOD performance, with ID chosen hyperparameters.

---

> > > > ### Author Response · Authors · 2023-11-21
> > > > **Response to Reviewer 6JHA - part 4**
> > > >
> > > > * ***Novelty detection algorithm***
> > > > Table 1 shows the impact of our Stylist on different novelty detection algorithms. It showcases a consistent improvement over baselines in almost all scenarios.
> > > > For our default novelty detection algorithm, which is kNN, we provide below an ablation regarding the choice of the number of neighbors (k). We highlight the robustness of the approach w.r.t. k, as the variance is in range [0.002, 0.004] for CLIP and in range [0.013, 0.160] for ResNet-18 features.
> > > >
> > > > |Dataset | Features | Optimal nr. of features selection approach | k | ROC-AUC all feat. | ROC-AUC Stylist feat. | % selected feat. |
> > > > |--|--|--|--|--|--|--|
> > > > |COCOShift_balanced | CLIP | Based on ID val set | 5 | 94.96 | 95.29 (+0.33) | 95  |
> > > > | | | | 10 | 95.07 | 95.38 (+0.31) | 95 |
> > > > | | | | 20 | 95.08 | 95.41 (+0.33) | 95 |
> > > > | | | | 30 | 95.06 | 95.40 (+0.34) | 95 |
> > > > | | | | **Variance w.r.t. k** | **0.003** | **0.003** | - |
> > > > | | | Based on OOD test set | 5 | 94.96 | 95.29 (+0.33) | 95  |
> > > > | | | | 10 | 95.07 | 95.38 (+0.31) | 95 |
> > > > | | | | 20 | 95.08 | 95.41 (+0.33) | 95 |
> > > > | | | | 30 | 95.06 | 95.40 (+0.34) | 95 |
> > > > | | | | **Variance w.r.t. k** | **0.003** | **0.003** | - |
> > > > | | | **Variance w.r.t. features selection approach** | - | -| **0** | - |
> > > > | | ResNet-18 | Based on ID val set | 5 | 81.51 | 83.58 (+2.07) | 10  |
> > > > | | | | 10 | 81.36 | 83.51 (+2.15) | 10 |
> > > > | | | | 20 | 81.12 | 83.36 (+2.24) | 10 |
> > > > | | | | 30 | 80.93 | 83.24 (+2.31) | 10 |
> > > > | | | | **Variance w.r.t. k** | **0.065** | **0.023**  | - |
> > > > | | | Based on OOD test set | 5 | 81.51 | 84.08 (+2.57)  | 40  |
> > > > | | | | 10 | 81.36 | 83.92 (+2.56) | 40 |
> > > > | | | | 20 | 81.12 | 83.68 (+2.56) | 40 |
> > > > | | | | 30 | 80.93 | 83.49 (+2.56) | 40 |
> > > > | | | | **Variance w.r.t. k** | **0.065** | **0.069** | - |
> > > > | | | **Variance w.r.t. features selection approach** | - |- |**0.068** |- |
> > > > |COCOShift75 | CLIP | Based on ID val set | 5 | 94.77 | 95.11 (+0.34) | 95  |
> > > > | | | | 10 | 94.88 | 95.22 (+0.34) | 95 |
> > > > | | | | 20 | 94.91 | 95.25 (+0.34) | 95 |
> > > > | | | | 30 | 94.88 | 95.21 (+0.33) | 95 |
> > > > | | | | **Variance w.r.t. k** | **0.004** | **0.003** | - |
> > > > | | | Based on OOD test set | 5 | 94.77 | 95.11 (+0.34) | 95  |
> > > > | | | | 10 | 94.88 | 95.22 (+0.34) | 95 |
> > > > | | | | 20 | 94.91 | 95.25 (+0.34) | 95 |
> > > > | | | | 30 | 94.88 | 95.21 (+0.33) | 95 |
> > > > | | | | **Variance w.r.t. k** | **0.004** | **0.003** | - |
> > > > | | | **Variance w.r.t. features selection approach** | - | - |**0** |- |
> > > > | | ResNet-18 | Based on ID val set | 5 | 81.10 | 84.96 (+3.86) | 10  |
> > > > | | | | 10 | 80.91 | 84.90 (+3.99) | 10 |
> > > > | | | | 20 | 80.62 | 84.79 (+4.17) | 10 |
> > > > | | | | 30 | 80.40 | 84.71 (+4.31) | 10 |
> > > > | | | | **Variance w.r.t. k** | **0.096** | **0.013** | - |
> > > > | | | Based on OOD test set | 5 | 81.10 | 84.96 (+3.86) | 10  |
> > > > | | | | 10 | 80.91 | 84.90 (+3.99) | 10 |
> > > > | | | | 20 | 80.62 | 84.79 (+4.17) | 10 |
> > > > | | | | 30 | 80.40 | 84.71 (+4.31) | 10 |
> > > > | | | | **Variance w.r.t. k** | **0.096** | **0.013** | - |
> > > > | | | **Variance w.r.t. features selection approach** | - | - | **0** | - |
> > > > |COCOShift95 | CLIP | Based on ID val set | 5 | 94.59 | 94.97 (+0.38) | 95  |
> > > > | | | | 10 | 94.65 | 95.02 (+0.37) | 95 |
> > > > | | | | 20 | 94.61 | 94.99 (+0.38) | 95 |
> > > > | | | | 30 | 94.53 | 94.92 (+0.39) | 95 |
> > > > | | | | **Variance w.r.t. k** | **0.002** | **0.002** | - |
> > > > | | | Based on OOD test set | 5 | 94.59 | 94.97 (+0.38) | 95  |
> > > > | | | | 10 | 94.65 | 95.02 (+0.37) | 95 |
> > > > | | | | 20 | 94.61 | 94.99 (+0.38) | 95 |
> > > > | | | | 30 | 94.53 | 94.92 (+0.39) | 95 |
> > > > | | | | **Variance w.r.t. k** | **0.002** | **0.002** | - |
> > > > | | | **Variance w.r.t. features selection approach** | - |- |**0** |- |
> > > > | | ResNet-18 | Based on ID val set | 5 | 80.72 | 85.53 (+4.81) | 25  |
> > > > | | | | 10 | 80.48 | 85.38 (+4.90) | 25 |
> > > > | | | | 20 | 80.11 | 85.16 (+5.05) | 25 |
> > > > | | | | 30 | 79.82 | 85.01 (+5.19) | 25 |
> > > > | | | | **Variance w.r.t. k** | **0.160** | **0.052** | - |
> > > > | | | Based on OOD test set | 5 | 80.72 | 85.66 (+4.94) | 30  |
> > > > | | | | 10 | 80.48 | 85.51 (+5.03) | 30 |
> > > > | | | | 20 | 80.11 | 85.29 (+5.18) | 30 |
> > > > | | | | 30 | 79.82 | 85.13 (+5.31) | 30 |
> > > > | | | | **Variance w.r.t. k** | **0.160** | **0.054** | - |
> > > > | | | **Variance w.r.t. features selection approach** | - | - | **0.008** | - |

---

> > > > > ### Author Response · Authors · 2023-11-21
> > > > > **Response to Reviewer 6JHA - part 5**
> > > > >
> > > > > **W5 Dataset contribution is marginal**
> > > > >
> > > > > **W5.1. (1) The necessity of COCOShift95 over prior works is unclear. The paper introduces a new benchmark called COCOShift95, where objects are cut and shifted to different backgrounds. Consequently, this benchmark is artificial and does not demand substantial effort, unlike previous works such as fMoW and DomainNet, which require considerable effort to collect realistic images. What new insights can be gained from COCOShift95, aside from adding yet another column to the table.**
> > > > >
> > > > > We built the COCOShift benchmark to check the impact spurious correlation has on the ND performance. To this extent, COCOShift brings a controllable level of spurious correlations (in the form of COCOShift balanced, COCOShift75, COCOShift90, and COCOShift95), unlike other currently available benchmarks. In section 4.1, we evaluate Stylist robustness to dataset spuriousness level, an analysis that could not be conducted on other benchmarks. By using COCOShift benchmark, we found out that Stylist performs effectively in both heavy spuriousness and little to no spuriousness scenarios. As opposed to this, existing datasets as fMoW and DomainNet (on which we report the main results of the work) were not suitable for doing the analysis on what kind of features our algorithm focuses on, in a disentangled or real life scenario (Fig. 6), since style and content could not be easily separated: a building picture must have a date, an image of an object must have a representation style (paint, real, drawing, etc), while in COCOShift, the object can be defined without a background.
> > > > >
> > > > >
> > > > > **W5.2. (2) The cut and paste strategy has been considered in prior works. The approach of creating an artificial dataset through cut and paste has been proposed in previous works, such as Waterbirds and Background Challenge. Additionally, why not use more natural benchmarks, such as MetaShift, which also include multi-domain images from COCO? Therefore, the technical contribution of the proposed dataset collection strategy is also not convincing.**
> > > > >
> > > > > We took them into account in the beginning, but each one has at least one major flaw that led us to discard them.
> > > > >
> > > > > ***[Waterbirds]*** This dataset has only 2 environments, one used for training, and one used for testing. The existence of multiple environments is essential to define what the style consists of, so we need to see at least 2 domains to determine the environment-biased features. Indeed, it is very similar to COCOShift in other regards.
> > > > >
> > > > > ***[Background challenge benchmark]*** This benchmark does not provide labels for the style, but only for the content, so we can’t run our approach on top of it as we need to know the environment for each sample. Furthermore, by construction, we have secondary objects in the samples, for which we don’t have labels, but they come from the same set of classes as the main content. This makes it impossible to have a clear separation between the style and content.
> > > > >
> > > > > ***[MetaShift]*** Here, the annotations of each image encompass nearly all the objects present within it. Usually, there are 3 or more different objects per image, which intersect the content and environment label sets. Also, the labels are very noisy. A clear example is the image highlighted at https://huggingface.co/datasets/metashift?row=3 . The image is labeled as a bus in a fence context, but the predominant objects in the foreground are the horses, which is a given label for other samples. This is such a common occurrence that we only found 2159 clean samples (with no conflicting annotations). For non-intersecting content and environment label sets, there are only 281 samples. We consider that these numbers are too small to create a reliable benchmark.

---

> > > > > > ### Comment · Reviewer_6JHA · 2023-12-04
> > > > > > **Response to the rebuttal**
> > > > > >
> > > > > > Thank you for the extensive time and effort on the rebuttal. While I appreciate the efforts, I believe the review should prioritize clarity and conciseness over quantity and experiments. For instance, are lengthy tables essential for delivering the core message? Focusing on the message, I think the rebuttal did not clearly address my initial concerns.
> > > > > >
> > > > > > **W1. Limited scope**
> > > > > >
> > > > > > The rebuttal claims that Robust Novelty Detection (ND) is an important problem in real-life scenarios. However, the specific use case is not clarified and, therefore, not convincing. Also, current experimental setups are rather artificial. I would find this claim more credible if the authors presented a specific real-life scenario and conducted experiments in that context.
> > > > > >
> > > > > > Not only the problem scope, but I also cannot agree that applying a feature selection approach to improve OOD detection performance on new experimental setups is a substantial technical contribution.
> > > > > >
> > > > > > **W4 and Q2. Hyperparameter selection**
> > > > > >
> > > > > > I was mainly discussing the percentage of selected features. Is this robust to diverse environment combinations? In real-life scenarios, there are numerous styles and content combinations. It's still not convincing that this method would be robust beyond synthetic benchmark combinations.
> > > > > >
> > > > > > **W5 Dataset contribution is marginal**
> > > > > >
> > > > > > I agree that prior benchmarks lack controllable spurious correlations, and creating a synthetic benchmark to test this could be meaningful. However, I still don't think this constitutes a substantial dataset contribution; rather, it seems like an inexpensive adjustment to an existing dataset to validate the hypothesis. I'm not saying this dataset is meaningless, but proposing it cannot be considered a significant contribution deserving of its own standalone bullet point in bold. Instead, it could be viewed as a minor contribution that can be briefly explained when highlighting the experimental results. This also aligns with my W1, that the contributions of ICLR papers should aim to be more substantial.

---

### Official Review · Reviewer_gWKH · 2023-10-29

**Soundness:** 3 good
**Presentation:** 3 good
**Contribution:** 2 fair
**Rating:** 5
**Confidence:** 4

**Summary:**

This paper considers a robust novelty detection problem of finding semantic changes while being robust to style distributional shifts. The authors propose a feature selection method, which ranks the features by evaluating the differences of features across domains to achieve invariance, thereby removing the influence of spurious correlations. In the experiments, the effectiveness of the proposal is validated where the feature selection improves the subsequent novel detection algorithms under distribution shift.

**Strengths:**

1. The problem presented is important and highly relevant to the machine learning community. The proposal is straightforward and technologically feasible.
2. The proposed feature selection can be combined with different novel detection algorithms and improve their robustness when facing distribution shifts. The authors also demonstrate that the proposal can effectively select environment-invariant features.
3. The overall presentation is well-organized and easy to follow.

**Weaknesses:**

Important references are missing. There have been some works on how to achieve novelty detection in the case of domain shift, which is the same as this paper. For example:
1. Open domain generalization with domain-augmented meta-learning. CVPR 2021
2. Open-set learning under covariate shift. ML 2022
3. Open-Set Single Domain Generalization. ICLR 2022
These works all consider open-set learning/novelty detection under distribution shift. Moreover, the ML'22 and ICLR’22 works could improve cross-domain generalization without the requirement of multiple domains.

The previous works mainly focus on improving model robustness from the perspective of representation learning, while the current work assumes that the representation has already been well learned and mainly focuses on feature selection.

How does the author consider the sufficiency of representation capacity? Can we assume that the representation has been sufficiently learned after thorough across-domain representation learning? Will it still learn spurious correlations? Comparison with these works, especially the specific clarification of the applicable scope for this work, will further improve this manuscript.

**Questions:**

1. Differences with existing work and the rationale behind the assumptions made in the current work. [See weakness part]
2. In practice, how to determine the proportion of selected features? As shown in Figure 3. An inappropriate number of selected features can lead to a drop in ROC-AUC.

---

> ### Author Response · Authors · 2023-11-21
> **Response to Reviewer gWKH - part 1**
>
> Thank you for the detailed analysis of our work. Below we address each of the raised concerns.
>
> **Q1 - W1.  Important references are missing. There have been some works on how to achieve novelty detection in the case of domain shift, which is the same as this paper. For example:**
>
> * **Open domain generalization with domain-augmented meta-learning. CVPR 2021**
>
> * **Open-set learning under covariate shift. ML 2022**
>
> * **Open-Set Single Domain Generalization. ICLR 2022**
>
> **These works all consider open-set learning/novelty detection under distribution shift. Moreover, the ML’22 and ICLR’22 works could improve cross-domain generalization without the requirement of multiple domains**
>
> This is a very important observation, we address it below (we refer to those methods in the revised paper).
>
> ***[open set vs novelty detection]*** Open-set (OS) and novelty detection (ND) are strongly related, but also different in some key aspects. OS requires more supervision than ND, as during training, OS has access to the semantic labels of the normal/known data. Meanwhile, ND observes during training the set of normal classes, without having access to their specific content labels. OS is typically approached in a supervised learning context, while ND methods are often employed in an unsupervised context.  The indicated papers are not an exception, requiring annotated semantic labels for the training set. CVPR’21 optimizes a meta-learning objective that requires semantic labels. ML’22 requires the target concepts in order to identify the semantic and environmental parts of the image for the EnvMix step. ICLR’22 employs semantic labels for the minimization stage and implicitly they are used for increasing the diversity of the source domain and the generation of auxiliary unknown class samples.
>
> ***[requirement of multiple domains]*** Our Stylist employs the multi-environment setup as a way of defining the style of the images, namely the covariate shifts that should be ignored. Our algorithm is flexible and not restricted to a fixed definition of style and content. Content should not be understood as elements of WordNet hierarchy, but rather as what is constant between environments. Similarly, we define the style as the elements that vary between environments. In this setup, the absence of multiple training domains means the absence of a defined problem.
>
> ***CVPR’21*** Open Domain Generalization with Domain-Augmented Meta-Learning. Yang Shu, Zhangjie Cao, Chenyu Wang, Jianmin Wang, Mingsheng Long, CVPR 2021
>
> ***ML’22*** Open-set learning under covariate shift. Jie-Jing Shao, Xiao-Wen Wang, Lan-Zhe Guo, Machine Learning 2022
>
> ***ICLR’22*** CrossMatch: Cross-Classifier Consistency Regularization for Open-Set Single Domain Generalization. Ronghang Zhu, Sheng Li, ICLR 2022
>
> **Q1 (and W2). The previous works mainly focus on improving model robustness from the perspective of representation learning, while the current work assumes that the representation has already been well learned and mainly focuses on feature selection.  How does the author consider the sufficiency of representation capacity? Can we assume that the representation has been sufficiently learned after thorough across-domain representation learning? Will it still learn spurious correlations? Comparison with these works, especially the specific clarification of the applicable scope for this work, will further improve this manuscript.**
>
> ***[sufficiency of representation capacity]*** We rely on the existence of a pre-trained representation that captures relevant features for both known and unknown content and style. Considering the availability of heavily pre-trained models and scenarios like the use case illustrated below, such a representation can be easily accessible. As we are focusing on a flexible definition of style and content, an ideal representation will evenly capture elements of style and content.
>
> ***[will it still learn spurious correlations]*** In the unsupervised setup of novelty detection, during training, we observe normal samples in a few training environments. Assuming a perfectly balanced dataset, the ND algorithm will immediately correlate the normal samples with the seen training environments as it does not have access to semantic labels, as in the case of open-set recognition, to guide the learning process towards semantic features. Also, we can deal with biased datasets, which can further enhance this effect. In this context, the feature ranking strategy employed by Stylist will ensure that the novelty detection algorithm only has access to the relevant features, while ignoring environment-biased ones.

---

> > ### Author Response · Authors · 2023-11-21
> > **Response to Reviewer gWKH - part 2**
> >
> > ***[use case]*** Camera algorithms that work on self-driving cars face lots of challenges regarding generalization properties. They need to be robust to various landscapes, building architectures, even car appearances, or other smaller particularities that can arise from one place to another. In a real-life scenario like this, getting ***a) semantic labels*** for samples are usually ***expensive***. Also, in this case, ***b) environment labels*** can be relatively ***cheap/free*** to get (e.g. group samples based on the location/city/region/country). Another important aspect is the ***c) existence of good pre-trained representation (that generalizes well)*** for your specific task. This is a good real-world use case example in which our Stylist can take advantage of a) and b) and improve c) the quality of the representation, by identifying some spurious features.
> >
> > **Q2 - In practice, how to determine the proportion of selected features? As shown in Figure 3. An inappropriate number of selected features can lead to a drop in ROC-AUC.**
> >
> > We can determine the optimal proportion of features following a validation methodology on the ID validation set or by employing an oracle strategy and interrogating the OOD test set. In the table below we compare these approaches, highlighting that the optimal number of features is stable between the two setups. The variance between the two approaches is <=0.068.  Also, we emphasize that Stylist manages to always improve over the baseline considering 100% of the features, in all the considered configurations.
> >
> > |Dataset | Features | Optimal nr. of features selection approach | ROC-AUC all feat. | ROC-AUC Stylist feat. | % selected feat. |
> > |--|--|--|--|--|--|
> > |COCOShift_balanced | CLIP | Based on ID val set | 95.06 | 95.40 (+0.34) | 95 |
> > | |  | Based on OOD test set | 95.06 | 95.40 (+0.34) | 95 |
> > | |  | **Variance w.r.t. features selection approach** | - | **0** | - |
> > | | ResNet-18 | Based on ID val set | 80.93 | 83.24 (+2.31) | 10 |
> > | |  | Based on OOD test set | 80.93 | 83.49 (+2.56) | 40 |
> > | |  | **Variance w.r.t. features selection approach** | - | **0.068** | - |
> > |COCOShift75 | CLIP | Based on ID val set | 94.88 | 95.21 (+0.33) | 95 |
> > | |  | Based on OOD test set | 94.88 | 95.21 (+0.33) | 95 |
> > | |  | **Variance w.r.t. features selection approach** | - | **0** | - |
> > | | ResNet-18 | Based on ID val set | 80.40 | 84.71 (+4.31) | 10 |
> > | |  | Based on OOD test set | 80.40 | 84.71 (+4.31) | 10 |
> > | |  | **Variance w.r.t. features selection approach** | - | **0** | - |
> > |COCOShift95 | CLIP | Based on ID val set | 94.53 | 94.92 (+0.39) | 95 |
> > | |  | Based on OOD test set | 94.53 | 94.92 (+0.39) | 95 |
> > | |  | **Variance w.r.t. features selection approach** | - | **0** | - |
> > | | ResNet-18 | Based on ID val set | 79.82 | 85.01 (+5.19) | 25 |
> > | |  | Based on OOD test set | 79.82 | 85.13 (+5.31) | 30 |
> > | |  | **Variance w.r.t. features selection approach** | - | **0.008** | - |

---

> > > ### Comment · Reviewer_gWKH · 2023-12-01
> > >
> > > Thank you for the detailed responses, which have partially addressed my concerns. However, I still feel that the assumptions made in the proposal are overly optimistic, so I will maintain the initial score.

---

### Official Review · Reviewer_Diep · 2023-10-31

**Soundness:** 3 good
**Presentation:** 3 good
**Contribution:** 2 fair
**Rating:** 5
**Confidence:** 3

**Summary:**

The paper addresses the problem of detecting novel environments in machine learning models, which is important for ensuring their robustness and reliability. The authors note that existing methods for novelty detection often rely on the assumption that the training and testing data come from the same distribution, which is not always the case in real-world scenarios. They also point out that previous work has focused on either semantic or style changes, but not both, and that there is a need for a method that can separate these two types of changes and detect novel environments based on semantic changes while ignoring style changes.

To address this problem, the authors propose a novel method called Stylist, which leverages pretrained embeddings to separate semantic and style changes and rank features based on their relevance for detecting novel environments. The authors note that their approach is different from previous work in that it focuses on feature ranking rather than domain adaptation or anomaly detection. They also highlight the importance of identifying environment-biased features that contain spurious correlations and should be ignored for novelty detection.

The authors' contributions include a formalization of the problem of detecting novel environments based on semantic and style changes, a feature ranking approach that focuses on dropping environment-biased features, and an evaluation of their method on several benchmark datasets. They also provide insights into the impact of pretrained feature extractors and the potential applications of their method beyond novelty detection.

**Strengths:**

1. One of the strengths of Stylist is its ability to identify environment-biased features that contain spurious correlations and should be ignored for novelty detection. This is an important contribution, as spurious correlations can lead to false positives and negatively impact the reliability of machine learning models. By removing these features, Stylist can improve the generalization performance of novelty detection algorithms and make them more robust to changes in the environment.

2. Another strength of Stylist is its potential for interpretability. By ranking features based on their relevance for detecting novel environments, Stylist can provide insights into which features are important for the task at hand and which ones are not. This can be useful for understanding the behavior of machine learning models and for identifying areas for improvement.

**Weaknesses:**

1.  While the authors' approach of leveraging pretrained embeddings to separate semantic and style changes is innovative, the overall contribution of the paper may not be novel enough to warrant publication in a top-tier conference or journal. The authors could strengthen their contribution by providing more evidence of the novelty of their approach and by comparing it with other state-of-the-art methods for novelty detection.

2.  While the authors have provided some results on synthetic and real-world datasets, the evaluation of their method could be more comprehensive. Specifically, the authors could provide more details on the experimental setup, such as the choice of hyperparameters and the number of trials, to ensure that their results are reproducible. Additionally, the authors could compare their method with other state-of-the-art methods for novelty detection to better understand its strengths and weaknesses.

3. The paper focuses on the problem of detecting novel environments based on semantic and style changes, but it does not address other important aspects of novelty detection, such as temporal changes or changes in the data distribution over time. The authors could expand the scope of their work to address these other aspects of novelty detection and provide a more comprehensive solution to the problem.

**Questions:**

1. The authors mention that their method focuses on feature ranking rather than domain adaptation or anomaly detection, but it is not clear how this approach is fundamentally different from other methods. Could the authors provide more details on how their method differs from existing methods and what makes it unique?

2. The authors could provide more details on the experimental setup, such as the choice of hyperparameters and the number of trials, to ensure that their results are reproducible. Additionally, the authors could compare their method with other state-of-the-art methods for novelty detection to better understand its strengths and weaknesses. Could the authors provide more details on the experimental setup and the comparison with other methods?

---

> ### Author Response · Authors · 2023-11-21
> **Response to Reviewer Diep - part 1**
>
> We thank the reviewer for the observations and for pointing out ways to improve the analysis of our work. Below we address each of your comments and we have included the additional ablations in the supplementary material.
>
> **W3. The paper focuses on the problem of detecting novel environments based on semantic and style changes, but it does not address other important aspects of novelty detection, such as temporal changes or changes in the data distribution over time. The authors could expand the scope of their work to address these other aspects of novelty detection and provide a more comprehensive solution to the problem.**
>
> This is a very interesting point raised by the reviewer. The proposed approach is flexible to the definition of style and content. Style should not be understood solely as characteristics that separate for example sketches, real images, or paintings (as in DomainNet), or specific background characteristics like water, sand, or forest (as in Waterbirds, or fMoW with regions environments), but rather as the general covariate shift that we wish to be robust to (as temporal shifts that you mentioned).
>
> Our Stylist method does not make any other assumption on what the style is, the only way to provide a definition for what is the style, is the multi-environment setup considered in our work: what is changing between environments is irrelevant for the final task, and we call it style. So, if our training environments are composed of aerial images of a city observed over a long period of time, the style will capture those changes that occur over time.
> Moreover, we did a test to emphasize your proposed temporal shift use-case. First, we applied Stylist for the first 5 years of fMoW. Next, we observed the ND performance using our solution on a later year (with a visible temporal shift). The ROC-AUC improvement is around 1%, when compared with using all the features, emphasizing that our method works for different types of styles, specified and embedded as supervision signals from the multi-environment labels setup.
>
>
> **Q1 (and W1) - The authors mention that their method focuses on feature ranking rather than domain adaptation or anomaly detection, but it is not clear how this approach is fundamentally different from other methods. Could the authors provide more details on how their method differs from existing methods and what makes it unique?**
>
> We have completed the comparative analysis from our manuscript tackling the following points:
>
> ***[Robust Novelty Detection (our task) vs. Anomaly Detection]*** While anomaly detection is focused on identifying any abnormal or previously unseen elements, robust novelty detection focuses on identifying content distribution changes, while ignoring style changes that may appear. Considering the first example from Figure 1 of our manuscript, in anomaly detection, the sketch of the banana should be identified as an anomaly, because the style change is still considered an anomaly. In Robust Novelty Detection, the banana is classified as a normal class, because we should be robust to style changes.
>
> ***[Robust Novelty Detection (our task) vs. Open Set Recognition]*** Open-set (OS) and novelty detection (ND) are strongly related, but also different in some key aspects. OS requires more supervision than ND, as during training, OS has access to the semantic labels of the normal / known data. Meanwhile, ND observes during training the set of normal classes, without having access to their specific content labels. OS is typically approached in a supervised learning context, while ND methods are often employed in an unsupervised context.
>
> ***[Stylist (ours) vs. Domain Adaptation]*** Stylist is rather a domain generalization technique, removing environment-biased features and in consequence building representations that are robust to covariate shifts. Different from a domain adaptation technique (like TENT or TTT-MAE), Stylist does not require access to unlabeled target environment (test) data in order to compute the feature ranking.

---

> ### Author Response · Authors · 2023-11-21
> **Response to Reviewer Diep - part 2**
>
> ***[Stylist (ours) vs. Domain Generalization Approaches]***
> ***Similar*** to domain generalization techniques, we work in a multi-environment setup. During training, we observe data from at least two different environments and we implicitly define the style of our data as the covariate shift observed between those training environments. This can be seen as a way of defining the shifts that we want to be robust to. This way we give flexibility to our algorithm, providing the signal for robustness only from the environment labels. ***Different*** from other generalization approaches like IRM and LISA, we do not require access to both environments and semantic labels for our in-distribution training data. Our Stylist only requires style labels in order to compute the feature ranking.
>
> ***TENT*** - Fully Test-Time Adaptation by Entropy Minimization. Dequan Wang, Evan Shelhamer, Shaoteng Liu, Bruno Olshausen, Trevor Darell,. ICLR 2021
>
> ***TTT-MAE*** - Test-Time Training with Masked Autoencoders. Yossi Gandelsman, Yu Sun, Xinlei Chen, Alexei Efros, NeurIPS 2022
>
> ***IRM*** - Invariant Risk Minimization: Martin Arjovsky, Léon Bottou, Ishaan Gulrajani, David Lopez-Paz
>
> ***LISA*** - Improving Out-of-Distribution Robustness via Selective Augmentation: Huaxiu Yao, Yu Wang, Sai Li, Linjun Zhang, Weixin Liang, James Zou, Chelsea Finn, ICML 2022
>
> **Q2 (and W2) - The authors could provide more details on the experimental setup, such as the choice of hyperparameters and the number of trials, to ensure that their results are reproducible... Could the authors provide more details on the experimental setup and the comparison with other methods?**
>
> ***[experimental setup - number of trials]*** Stylist does not involve a random component and its results are reproducible given the same set of hyperparameters, as discussed below. For clarity, we have attached the code.
>
> ***[experimental setup - hyperparameters]*** Stylist is robust to the choice of hyperparameters. We emphasize the constant improvement illustrated below.
>
> Hyperparameters:
>
> * ***The feature extractor:***
> In Table 2 of our manuscript we provide an analysis of the performance w.r.t. the considered feature extractor. Our Stylist, improves the performance regardless of the selected feature extractor.
>
> * ***The distance metric*** and the ***combination between pairwise distances*** employed in Step1 of our algorithm:
> In Figure 7 of our manuscript we provide an ablation study regarding this matter. We highlight the performance improvement of our Stylist, irrespective of the considered configuration.
>
> * ***The percent of selected features***, for Step2 of our algorithm:
> As highlighted in Figure 3, Stylist consistently improves over the baseline w.r.t. the percent of considered features, proving that the provided feature ranking is relevant for the novelty detection problem. To select an optimal percent of features per setup, we employ a validation step, analyzing either the performance on an ID validation set or the performance on an OOD test set. As highlighted in the table below, there is a very small performance variance between those techniques (< 0.07 for ResNet-18 features, and even 0 for CLIP features), highlighting that the performance is stable. This is an important property of our algorithm, managing to improve OOD performance, with ID chosen hyperparameters.

---

> ### Author Response · Authors · 2023-11-21
> **Response to Reviewer Diep - part 3**
>
> * ***Novelty detection algorithm***
> Table 1 shows the impact of our Stylist on different novelty detection algorithms. It showcases a consistent improvement over baselines in almost all scenarios.
> For our default novelty detection algorithm, which is kNN, we provide below an ablation regarding the choice of the number of neighbors (k). We highlight the robustness of the approach w.r.t. k, as the variance is in range [0.002, 0.004] for CLIP and in range [0.013, 0.160] for ResNet-18 features.
>
>
> |Dataset | Features | Optimal nr. of features selection approach | k | ROC-AUC all feat. | ROC-AUC Stylist feat. | % selected feat. |
> |--|--|--|--|--|--|--|
> |COCOShift_balanced | CLIP | Based on ID val set | 5 | 94.96 | 95.29 (+0.33) | 95  |
> | | | | 10 | 95.07 | 95.38 (+0.31) | 95 |
> | | | | 20 | 95.08 | 95.41 (+0.33) | 95 |
> | | | | 30 | 95.06 | 95.40 (+0.34) | 95 |
> | | | | **Variance w.r.t. k** | **0.003** | **0.003** | - |
> | | | Based on OOD test set | 5 | 94.96 | 95.29 (+0.33) | 95  |
> | | | | 10 | 95.07 | 95.38 (+0.31) | 95 |
> | | | | 20 | 95.08 | 95.41 (+0.33) | 95 |
> | | | | 30 | 95.06 | 95.40 (+0.34) | 95 |
> | | | | **Variance w.r.t. k** | **0.003** | **0.003** | - |
> | | | **Variance w.r.t. features selection approach** | - | -| **0** | - |
> | | ResNet-18 | Based on ID val set | 5 | 81.51 | 83.58 (+2.07) | 10  |
> | | | | 10 | 81.36 | 83.51 (+2.15) | 10 |
> | | | | 20 | 81.12 | 83.36 (+2.24) | 10 |
> | | | | 30 | 80.93 | 83.24 (+2.31) | 10 |
> | | | | **Variance w.r.t. k** | **0.065** | **0.023**  | - |
> | | | Based on OOD test set | 5 | 81.51 | 84.08 (+2.57)  | 40  |
> | | | | 10 | 81.36 | 83.92 (+2.56) | 40 |
> | | | | 20 | 81.12 | 83.68 (+2.56) | 40 |
> | | | | 30 | 80.93 | 83.49 (+2.56) | 40 |
> | | | | **Variance w.r.t. k** | **0.065** | **0.069** | - |
> | | | **Variance w.r.t. features selection approach** | - |- |**0.068** |- |
> |COCOShift75 | CLIP | Based on ID val set | 5 | 94.77 | 95.11 (+0.34) | 95  |
> | | | | 10 | 94.88 | 95.22 (+0.34) | 95 |
> | | | | 20 | 94.91 | 95.25 (+0.34) | 95 |
> | | | | 30 | 94.88 | 95.21 (+0.33) | 95 |
> | | | | **Variance w.r.t. k** | **0.004** | **0.003** | - |
> | | | Based on OOD test set | 5 | 94.77 | 95.11 (+0.34) | 95  |
> | | | | 10 | 94.88 | 95.22 (+0.34) | 95 |
> | | | | 20 | 94.91 | 95.25 (+0.34) | 95 |
> | | | | 30 | 94.88 | 95.21 (+0.33) | 95 |
> | | | | **Variance w.r.t. k** | **0.004** | **0.003** | - |
> | | | **Variance w.r.t. features selection approach** | - | - |**0** |- |
> | | ResNet-18 | Based on ID val set | 5 | 81.10 | 84.96 (+3.86) | 10  |
> | | | | 10 | 80.91 | 84.90 (+3.99) | 10 |
> | | | | 20 | 80.62 | 84.79 (+4.17) | 10 |
> | | | | 30 | 80.40 | 84.71 (+4.31) | 10 |
> | | | | **Variance w.r.t. k** | **0.096** | **0.013** | - |
> | | | Based on OOD test set | 5 | 81.10 | 84.96 (+3.86) | 10  |
> | | | | 10 | 80.91 | 84.90 (+3.99) | 10 |
> | | | | 20 | 80.62 | 84.79 (+4.17) | 10 |
> | | | | 30 | 80.40 | 84.71 (+4.31) | 10 |
> | | | | **Variance w.r.t. k** | **0.096** | **0.013** | - |
> | | | **Variance w.r.t. features selection approach** | - | - | **0** | - |
> |COCOShift95 | CLIP | Based on ID val set | 5 | 94.59 | 94.97 (+0.38) | 95  |
> | | | | 10 | 94.65 | 95.02 (+0.37) | 95 |
> | | | | 20 | 94.61 | 94.99 (+0.38) | 95 |
> | | | | 30 | 94.53 | 94.92 (+0.39) | 95 |
> | | | | **Variance w.r.t. k** | **0.002** | **0.002** | - |
> | | | Based on OOD test set | 5 | 94.59 | 94.97 (+0.38) | 95  |
> | | | | 10 | 94.65 | 95.02 (+0.37) | 95 |
> | | | | 20 | 94.61 | 94.99 (+0.38) | 95 |
> | | | | 30 | 94.53 | 94.92 (+0.39) | 95 |
> | | | | **Variance w.r.t. k** | **0.002** | **0.002** | - |
> | | | **Variance w.r.t. features selection approach** | - |- |**0** |- |
> | | ResNet-18 | Based on ID val set | 5 | 80.72 | 85.53 (+4.81) | 25  |
> | | | | 10 | 80.48 | 85.38 (+4.90) | 25 |
> | | | | 20 | 80.11 | 85.16 (+5.05) | 25 |
> | | | | 30 | 79.82 | 85.01 (+5.19) | 25 |
> | | | | **Variance w.r.t. k** | **0.160** | **0.052** | - |
> | | | Based on OOD test set | 5 | 80.72 | 85.66 (+4.94) | 30  |
> | | | | 10 | 80.48 | 85.51 (+5.03) | 30 |
> | | | | 20 | 80.11 | 85.29 (+5.18) | 30 |
> | | | | 30 | 79.82 | 85.13 (+5.31) | 30 |
> | | | | **Variance w.r.t. k** | **0.160** | **0.054** | - |
> | | | **Variance w.r.t. features selection approach** | - | - | **0.008** | - |

---

> > ### Author Response · Authors · 2023-11-21
> > **Response to Reviewer Diep - part 4**
> >
> > **Q2 (and W2) - ... Additionally, the authors could compare their method with other state-of-the-art methods for novelty detection to better understand its strengths and weaknesses. Could the authors provide more details on the experimental setup and the comparison with other methods?**
> >
> > ***[SOTA for Novelty Detection]*** Our method is for a) feature selection, while the downstream task is b) Novelty Detection, seen in the setup where the data is shifting.
> >
> > ***b)*** Indeed, we do not compare against SOTA solutions for Novelty Detection, since they have no focus on the distribution shift of the input data, unlike in our case, and the comparison would be pointless. The most similar setup we found for ND in out-of-distribution, is the Out-of-distribution detection, because there is a shift in data and for a good performance, the model should be able to detect it. So we compare against the SOTA for this task (Out-of-distribution detection), called kNN+. We also add to the comparison with classic Novelty Detection solutions, that could be easily adapted for our setup (see Tab. 1).
> >
> > ***a)*** Consequently, we also compared against other feature selection approaches, basic and adapted for multiple environments, env-aware (see Fig. 6).

---

### Official Review · Reviewer_guaR · 2023-11-01

**Soundness:** 3 good
**Presentation:** 2 fair
**Contribution:** 2 fair
**Rating:** 3
**Confidence:** 4

**Summary:**

This paper focuses on novelty detection studies. Specifically, the authors want to improve the robustness of the novelty detection method to the spurious relations. To this end, the authors a new feature selection method called Stylist. The method ranks features according to the distance between different environments and selects features according to the rank.

**Strengths:**

1) The idea is technically sound. The proposal first rank features according to their distance between environments and remove features responsible for spurious correlations to improve the robustness.

**Weaknesses:**

1) It is not clear how to compute the Eq.(1) and Eq.(2). For example, how to construct different environments? How to split the features extracted from a pre-trained model into $N$ dimensions? It seems that if the pre-trained feature is very large, the computational cost is expensive.
2) In the experiments, the authors simply compare the proposal with the feature selection method and novelty detection method. However, there are some algorithms related to spurious relations that can be easily adapted to solve the novelty detection problem. Moreover, the adopted novelty detection methods are not SOTA. So, the experiment results are not convincing. More related methods should be discussed.
3) In my view, the spurious problems can be easily addressed with data augmentation. So I think maybe there exist easier methods to solve the problem in this paper.

**Questions:**

As discussed above.

---

> ### Author Response · Authors · 2023-11-21
> **Response to Reviewer guaR - part 1**
>
> We thank the reviewer for the observations, we tried to better explain the raised points and will update them in the paper to make it easier to follow.
>
> **W1. How to construct different environments? How to split the extracted features into N dimensions? The computational cost is expensive.**
>
> ***[environments]*** For the current version of our method, we need to have labels for the environments, so we don’t build them, we consider them as given. In the generalization field, there are a lot of works that are based on having annotations for multiple training distributions (=our environments), like IRM, LISA, or all the other methods that report on the WILDS benchmark.
>
> ***[extracted features]*** We think there is a misunderstanding here, we don’t split the features into N dimensions, but simply take the entire pre-trained features (e.g. ResNet18, with 512 features) and decide for each feature (of all 512 features), with our method, how much it varies between different environments.
>
> ***[computational cost]*** Quite the opposite, the computational impact is really low. For each feature, we only need to compute the inter-environments Wasserstein distances, which is extremely fast (for instance, for 10k samples, split into 5 environments, it takes ~ 4 seconds on a single CPU, with no optimizations whatsoever, since this was not our focus). All those operations can also be easily parallelized on CPUs. For more details on the implementation, we also attach the code to the submission.
>
> ***IRM*** - Invariant Risk Minimization: Martin Arjovsky, Léon Bottou, Ishaan Gulrajani, David Lopez-Paz
>
> ***LISA*** - Improving Out-of-Distribution Robustness via Selective Augmentation: Huaxiu Yao, Yu Wang, Sai Li, Linjun Zhang, Weixin Liang, James Zou, Chelsea Finn
>
> ***WILDS*** - https://wilds.stanford.edu/datasets/
>
> &nbsp;
>
> **W2. Comparison with other spurious algorithms and SOTA for Novelty Detection. More related methods should be discussed.**
>
> ***[Comparison with invariant learning (or other spurious-related algorithms)]***
> There is a very important difference in the setup between the existing methods for learning features that are invariant to the environment (like IRM) and our Stylist:
>
> ***a)*** For computing the invariant features, all the others need both semantic (content) labels (namely content classes) and environment labels, while our Stylist only needs the environment labels. We argue that it is a huge effort to have those semantic labels, similar to the supervised vs unsupervised differences.
>
> ***b)*** Indeed, to continue the argument, one may say that we use pre-trained models. And this means we also need labels at some point. So, to balance the need for semantic labels, we can make the IRM also use pre-trained models this way: one can learn IRM to predict labels that a pre-trained method gives for the current dataset. But this is a tricky step, because it might result in a lot of knowledge loss, since for training the IRM, you will go beyond the pre-trained features, to a final target class that is not necessarily mapped over the new dataset. This way, the final supervision will greatly influence the representations, especially if there is not a 100% alignment between your new dataset and the pre-trained model training dataset (and this is a common case).
>
> ***c)*** Nevertheless, since semantic labels are available in our used datasets, we did an experiment that focuses on the complementarity between the methods, not on their differences (we explained before why this direct comparison is not fair): We first trained IRM on those semantic labels. Next, we applied Stylist on top of them. Interestingly, we see a 1% increase over using 100% of the features, showing that the two methods are somehow complementary.
>
> ***IRM*** - Invariant Risk Minimization: Martin Arjovsky, Léon Bottou, Ishaan Gulrajani, David Lopez-Paz

---

> ### Author Response · Authors · 2023-11-21
> **Response to Reviewer guaR - part 2**
>
> ***[SOTA for Novelty Detection]*** Our method is for ***a) feature selection***, while the downstream task is ***b) Novelty Detection***, analysed in the setup where the data is shifting.
>
> ***b)*** Indeed, we do not compare against SOTA solutions for Novelty Detection, since they have no focus on the distribution shift of the input data, unlike in our case, and the comparison would be pointless. The most similar setup we found for ND in out-of-distribution, is the Out-of-distribution detection, because there is a shift in data and for a good performance, the model should be able to detect it. So we compare against the SOTA for this task (Out-of-distribution detection), called kNN+. We also add to the comparison with classic Novelty Detection solutions, that could be easily adapted for our setup (see Table 1).
>
> ***a)*** Consequently, we also compared against other feature selection approaches, basic and adapted for multiple-environments (env-aware) (see Figure 6).
>
> ***kNN+*** - Out-of-Distribution Detection with Deep Nearest Neighbors: Yiyou Sun, Yifei Ming, Xiaojin Zhu, Yixuan Li
>
>
> ***[More related methods]*** The reviewer does not specify exactly the related methods to be discussed. We add in the revised paper **Open-Set methods** and **Domain adaptation** ones (see related work). We have discussed already in the experimental section **Feature selection** and **Novelty detection** algorithms, and in the related work, **spurious/invariant learning** methods.
>
> &nbsp;
>
> **W3. The spurious problems can be easily addressed with data augmentation**
> Since it might be true that with the proper augmentations for each task and dataset, you might address some of the bias problems (like ImageNet has a bias that the main objects are always in the center), others can’t be addressed by this, e.g.:
>
> * If cows are always on the grass background in training, how can one augment the data to prepare it for other kinds of backgrounds? Just by first identifying the exact problem and building the exact custom augmentations on synthetic data (if you can for instance do a proper instance segmentation). But this is way more complicated and hand-designed than our proposed solution.
>
> * Another case is when the bias is distributed across the image, like CT image devices in hospitals that have different acquisition parameters that influence the entire image. This makes it impossible to augment in a generic way, since you don’t know how data will look for the next hospital. So you will need to see the “test” data. With our Stylist approach, you might figure out the features that are responsible for the acquisition parameters beforehand.
>
> More, please note that the best augmentations to make for a scenario, are task and data-dependent, as shown in ERM w/ targeted aug. Also, please take into account that there is a whole subfield of ML research regarding spurious correlations and biases.
>
> ***ERM w/ targeted aug***: Out-of-Distribution Robustness via Targeted Augmentations: Irena Gao, Shiori Sagawa, Pang Wei Koh, Tatsunori Hashimoto, Percy Liang

---

### Author Response · Authors · 2023-11-22
**Thank you for the review**

Dear reviewers,

Thank you for taking the time to thoroughly analyze our paper and for providing useful insights and suggestions that helped us improve our work.

We tried to address all the raised concernes in the rebuttal. We kindly ask you to revise our answers and let us know if there is anything else we can clarify.

Thank you again for your time and interest, we are eagerly waiting for your updates!

---

### Meta-Review · Area_Chair_mrUn · 2023-12-11

**Metareview:**

The paper considers the problem of robust novelty detection where the goal is to find novel semantic or content changes while being robust to style shifts. The proposed methods works by ranking pretrained features using Wasserstein distance across environments and removing features with highest Wasserstein distances. The method is motivated by the assumption that feature space is comprehensive enough to capture relevant distribution changes, and the style changes drive larger shifts in the feature space than content changes. Reviewers are of the view that the assumptions underlying the method are overly optimistic, and contribution of the paper is limited in scope and technical merit / novelty. Unfortunately these concerns make the paper unsuitable for publication at ICLR in its current form.

**Justification For Why Not Higher Score:**

Limited technical contribution and novelty; overly optimistic assumptions behind the proposed method.

**Justification For Why Not Lower Score:**

N/A

---

### Decision · Program_Chairs · 2024-01-16

Reject